# Ribosomal DNA copy number variation associates with hematological profiles and renal function in the UK Biobank

## Graphical abstract

## Authors

Francisco Rodriguez-Algarra,
David M. Evans, Vardhman K. Rakyan

## Correspondence

v.rakyan@qmul.ac.uk

## In brief

Rodriguez-Algarra et al. present an analysis of rDNA copy number (CN) variation in the UK Biobank (UKB), revealing associations with altered counts of specific blood cell subtypes, such as neutrophils, and estimated glomerular filtration rate, a marker of kidney function.

## Highlights

- Identification of ribosomal DNA CN variation in ~500,000 individuals

- rDNA CN does not associate with genetic variation elsewhere in the genome

- rDNA CN is associated with blood cell composition and markers of kidney function

- Associations are not due to reverse causality or common confounders

Rodriguez-Algarra et al., 2024, Cell Genomics 4, 100562
June 12, 2024 © 2024 The Author(s). Published by Elsevier Inc.

CellPress

## Article

# Ribosomal DNA copy number variation associates with hematological profiles and renal function in the UK Biobank

Francisco Rodriguez-Algarra,[1] David M. Evans,[2,3,4] and Vardhman K. Rakyan[1,5,*]

[1]The Blizard Institute, School of Medicine and Dentistry, Queen Mary University of London, London E1 2AT, UK
[2]Institute for Molecular Bioscience, The University of Queensland, Brisbane, QLD 4072, Australia
[3]Frazer Institute, The University of Queensland, Brisbane, QLD 4102, Australia
[4]MRC Integrative Epidemiology Unit, University of Bristol, Bristol BS8 2BN, UK
[5]Lead contact
*Correspondence: v.rakyan@qmul.ac.uk

## SUMMARY

The phenotypic impact of genetic variation of repetitive features in the human genome is currently under-studied. One such feature is the multi-copy 47S ribosomal DNA (rDNA) that codes for rRNA components of the ribosome. Here, we present an analysis of rDNA copy number (CN) variation in the UK Biobank (UKB). From the first release of UKB whole-genome sequencing (WGS) data, a discovery analysis in White British individuals reveals that rDNA CN associates with altered counts of specific blood cell subtypes, such as neutrophils, and with the estimated glomerular filtration rate, a marker of kidney function. Similar trends are observed in other ancestries. A range of analyses argue against reverse causality or common confounder effects, and all core results replicate in the second UKB WGS release. Our work demonstrates that rDNA CN is a genetic influence on trait variance in humans.

## INTRODUCTION

Genome-scale analyses have revolutionized our knowledge of the genetic architecture of common complex human traits. However, in most cases only a small proportion of the genetic component of trait variation has thus far been explained. Although rare genetic variation in the single-copy portion of the genome will undoubtedly be a major contributor to the remaining genetic component, a key limitation of most previous genetic association studies is that repetitive/multi-copy features have been ignored. This is now beginning to change with the recent availability of large-scale whole-genome sequencing (WGS)-based datasets, enabling more powerful and systematic investigations of how genetic variation within repetitive/multi-copy genomic features—such as variable nucleotide tandem repeats,[1] short tandem repeats,[2] telomeres,[3] and mitochondrial DNA[4]—contribute to human traits.

Another key repetitive genomic feature is the multi-copy, multi-locus 47S ribosomal DNA (rDNA) that codes for the 18S, 5.8S, and 28S ribosomal RNAs (rRNAs), thereby playing a central role in cellular function (Figure 1A). The 47S rDNA displays substantial inter-individual genetic variation in the form of CN (∼100–600 copies) and single-nucleotide variants (SNVs) in human populations.[5,6] In fact, rDNA genetic variation is found in most, if not all, species. The consequences of genetic variation within the rDNA are better understood in non-mammalian organisms, such as bacteria, in which specific copies of rDNA influ-

ence the stress response,[7,8] yeast, in which rDNA CN is thought to influence lifespan,[9] and *Drosophila*, in which rDNA CN may influence gene expression in the rest of the genome.[10] Thus far, studies that have attempted to study associations between rDNA genetic variation and human non-malignant phenotypes have been hampered by small sample sizes,[11–13] and/or profiling methods that are likely semi-quantitative.[14,15] Although associations between rDNA genetic variation and phenotype were found in some cases, the statistical evidence was not robust, at least by current GWAS standards. The only study to date that used sample numbers similar to those typically considered in recent GWASs analyzed rDNA CN variation in 7,268 individuals but did not find an association with autism spectrum disorder, the only phenotype that was studied.[16] Although robust rDNA CN changes have been reported for cancer, these are almost certainly a consequence of the cancer state, without any clear genetic evidence of being involved in the pathogenesis.[17,18] Therefore, strong evidence for rDNA-associated genetic variation impacting human traits is, as yet, lacking.

rDNA is not represented on currently available commercial array or exome capture platforms. Methods that do allow quantitative analysis of rDNA genetic variation in large sample numbers include digital-droplet PCR and high-depth WGS. Recently, the UK Biobank (UKB)—a population-based cohort of approximately half a million participants associated with a large amount of phenotypic and genomic data—released ∼30× coverage WGS data for a total of ∼490,505 participants

*(legend on next page)*

in two separate releases (200,004 in November 2021 and 290,501 in December 2023).[20–22] Although most diseases are under-represented in the UKB, and the majority of individuals are of White British (WB) ancestry, the UKB WGS dataset is still an unprecedented opportunity to study the potential influence of rDNA genetic variation on human phenotypes. Here, we present a systematic analysis of rDNA CN variation, and its association with common human traits, in the UKB.

## RESULTS

### Core influences on rDNA CN calls in the UKB

To provide a manageable focus for our study, and given evidence of genetic variation from previous literature,[5,6] we limited analyses to 47S rDNA CN. The UKB WGS data are provided pre-aligned to the Hg38 reference assembly, and re-aligning the data to a tailored assembly that includes a 47S rDNA consensus sequence is unfeasible in terms of cost and time.[5,7,16,19] We therefore developed a proxy estimation method for 47S rDNA CN that leverages various rDNA analogues present as pseudo-copies and unplaced contigs in Hg38, particularly from the highly conserved 18S subunit (Figure S1; STAR Methods). Validation in 94 WB individuals from the 1000 Genomes Project yielded highly correlated rDNA CN estimates between the proxy and tailored assembly methods (Figure 1B; Pearson's R = 0.97, $p = 6.5 \times 10^{-56}$), and absolute 47S rDNA copies per individual derived from these proxy estimates on UKB WB individuals are within the ranges previously reported in the 1000 Genomes Project[16] (Figure 1C). In addition, we and others have previously shown that WGS-based rDNA CN estimates correlate strongly with those derived from orthogonal ddPCR measurements.[5,19]

A potential technical influence on WGS-based rDNA CN calls is sequencing center-associated effects.[16] The first release of UKB WGS data was generated in three batches at two different sequencing centers: "Sanger Vanguard," "Sanger," and "deCODE." Indeed, we found that Sanger Vanguard and Sanger estimates display a higher mean relative to deCODE estimates (Figure 1D). Therefore, for subsequent analyses we either combined the Sanger releases and performed analyses on the Sanger and deCODE subsets separately, used rDNA CN estimates adjusted for sequencing center (Figure S2; STAR Methods), or included sequencing center alongside assessment

center, genetic principal components, and telomere length (given the sub-telomeric location of 47S rDNA) as covariates. Age and sex were also included as covariates, even though they did not show any association with rDNA CN (Figures S3–S5; ANOVA $p = 0.203$ for age, $p = 0.219$ for age squared; sex, Figures S3–S6; ANOVA $p = 0.839$). The other major influence on rDNA CN was self-reported ancestry (Figure 1E; ANOVA $p < 10^{-300}$). In particular, "Black or Black British" individuals displayed higher rDNA CN relative to other ancestries, consistent with previous reports.[6] Therefore, for the discovery analyses, we focused on a single ancestry, namely WB individuals, who comprise the vast majority of the WGS data (>85%, N = 169,919 in the first release, of which 157,227 remain after quality control; see STAR Methods).

### rDNA CN shows strong familial correlations, but is not associated with genetic variation elsewhere in the genome

Pairwise comparisons in monozygotic (MZ) twins, first-, second-, and third-degree relatives with both individuals sequenced in the same sequencing center batch showed highly significant correlations, and an expected decrease in Pearson's R with distance, from 0.95 to 0.11 (Figure 1F; using sequencing center-adjusted estimates on all pairs yields similar results; Figure S7). The data also further emphasize the technical robustness of the proxy rDNA CN calls.

We then asked if rDNA CN is influenced by genetic variation elsewhere in the genome and performed a GWAS of rDNA CN. Since the WGS data were generated from whole blood, we included principal components obtained from proportions of nucleated blood cell subtypes as covariates to control for the possibility of somatic differences in rDNA CN (reviewed by Hall et al.[23]; Figure S8). From over 7 million variants from 1000 Genomes Project-imputed genotypes at MAF > 1%, a GWAS of all WB participants yielded a single variant (rs62153030) at whole-genome significance level (beta = $2.64 \times 10^{-6}$, standard error [SE] = $4.77 \times 10^{-7}$, $p = 3.4 \times 10^{-8}$; Figure S9A; genomic inflation factor $\lambda_m = 1$; GWAS catalog accession GCST90356215). Identical results were obtained when blood cell composition principal components were not included (Figure S9B; GCST90356216). When considering only unrelated individuals (N = 127,231), no variants reached whole-genome significance (Figure 1G; GCST90356217), yielding

---

**Figure 1. Estimation of 47S rDNA copy number in the UKB**

(A) Schematic representation of the human rDNA loci on a cytogenetic ideogram of the Hg38 assembly as generated by the NIH's Genome Decoration Page, and the localization of the rRNAs in the Large Subunit (LSU) and Small Subunit (SSU) of the ribosome. Blue-tinted chromosomal segments indicate "variable regions," which are not necessarily related with rDNA despite overlapping the 47S clusters.

(B) Pearson's correlation between the total 47S rDNA CN values obtained with the previously published method employed by Rodriguez-Algarra et al.[19] and the proxy estimates obtained with the method proposed here ("18S Ratio") for 94 samples from the GBR population of the 1000 Genomes Project.

(C) Wilcoxon signed-rank test for the difference of means between previously published total rDNA CN estimates from the 1000 Genomes Project GBR population[16] and those we derive from the 18S Ratios of UKB White British (WB) participants from the first WGS release.

(D) Wilcoxon signed-rank tests for the difference of means between the 18S Ratios calculated on each of the UKB first WGS release sequencing center batches. Note, several previous studies have shown that 18S-based rDNA CN estimates correlate strongly with those derived from 28S or 5.8S.[5,6,16,19]

(E) Comparison of mean 18S Ratio and corresponding 95% confidence interval among top-level self-reported ethnic backgrounds in the UKB's first WGS release, split by sequencing center.

(F) Pearson's correlation between 18S Ratios in genetically identified relative pairs according to KING kinship thresholds[20] sequenced in the same sequencing center batch of the first WGS release.

(G) Manhattan plot for a GWAS of 18S Ratio in N = 127,231 unrelated WB participants using 1000 Genomes Project-imputed variants at MAF > 1%. The red dashed line indicates the minus $\log_{10} p$ value corresponding to a $5 \times 10^{-8}$ significance threshold.

**Figure 2. Association between rDNA CN and blood cell composition**

(A) Effect size (left) and significance level (right) for 18S Ratio associations from a phenome-wide screen in WB UKB participants from the first WGS release. Displayed phenotypes reach FDR < 0.01 for *N* = 157,227 WB participants and are measured in at least half of those samples, with >200 cases in case/control variables.

an SNP heritability coefficient of $h_g^2 = 0.000288 \pm 0.00513$. Furthermore, additional analyses, including conflicting results between sequencing centers, strongly indicated that the single variant identified above was a false positive (Figures S9C–S9E; GCST90356218 for Sanger and GCST90356219 for deCODE). This conclusion is reinforced by the results obtained in the second release UKB WGS data (presented below). Therefore, we find no robust evidence for genetic variation elsewhere in the genome influencing rDNA CN.

## Phenome-wide association screen of rDNA CN

To identify human traits genetically associated with rDNA CN, we performed a phenome-wide association screen of 2,722 phenotypes in the WB individuals, yielding 23 associations at FDR < 0.01 (Figure 2A; Table S1). No FDR-significant associations were obtained on a control analysis using permuted rDNA CN values (Table S2), and the directionality of effects was consistent for all 23 associations in sequencing center-specific screens (Figure S10; Tables S3 and S4). However, phenome-wide screens can only provide a broad overview of potential phenotypic associations. More detailed and focused analyses, with appropriate statistical models and covariates informed by domain knowledge, are necessary to ascertain the true nature of the potential relationships. Given that 20 of the 23 hits were with phenotypes classified as either "blood count" or "blood biochemistry" in the UKB, we proceeded to perform more comprehensive analyses focusing on these two UKB categories separately.

In the blood count category, the strongest associations included lymphocyte percentage (beta = −0.035 [95% confidence interval, −0.04 to −0.03], $p = 9.52 \times 10^{-43}$, FDR = $7.85 \times 10^{-40}$), neutrophil counts (beta = −0.033 [0.028–0.038], $p = 2.73 \times 10^{-36}$, FDR = $2.57 \times 10^{-33}$), and neutrophil percentage (beta = −0.032 [0.026–0.037], $p = 8.86 \times 10^{-34}$, FDR = $2.65 \times 10^{-31}$), all with N = 157,227 participants. We therefore posited rDNA CN associates with specific blood cell subtype combinations. Given the composite nature of blood and the shared mea-

surement procedure for the distinct cell types, we fit a multivariate linear model including all blood subtype counts to control for their potential mutual influence, yielding various putative positive and negative associations, most notably the positive association with neutrophil counts (Figure S11; N = 157,227, standardized beta (β) = 0.0273 [0.0221–0.0326], $p = 1.67 \times 10^{-24}$). rDNA CN thus associates with the abundance of multiple blood cell subtypes simultaneously. This combination of positive and negative associations with various blood subtypes is a feature of rDNA CN, as all pairwise correlations between blood cell subtype counts in the UKB are in fact positive when not accounting for rDNA CN (except for erythrocytes with platelets and basophils; Figure S12). The specific blood cell composition profile associated with rDNA CN is also highly reminiscent of well-established markers of systemic inflammation—such as neutrophil-to-lymphocyte ratio (NLR), platelet-to-lymphocyte ratio (PLR), and systemic immune-inflammation index (SII) (N× L/P). These markers also associate with rDNA CN: $\beta_{SII} = 0.0316$ [0.0265–0.0367], $p_{SII} = 9.91 \times 10^{-34}$, $\beta_{NLR} = 0.0266$ [0.0215–0.0317], $p_{NLR} = 1.55 \times 10^{-24}$; $\beta_{PLR} = 0.0201$ [0.0150–0.0252], $p_{PLR} = 7.8 \times 10^{-15}$ (all with N = 157,195; Figure 2B).

## The association between rDNA CN and blood cell composition is not due to reverse causality or confounder effects

In large-scale association studies of variation in the single-copy genome, it is assumed that the direction of causality is, either directly or indirectly, from the variant to the trait. rDNA CN is repetitive, however, and such genomic features have the potential to be somatically variable. The phenome-wide screen presented above thus does not, in isolation, reveal the direction of the association between rDNA CN and blood cell subset proportions. Therefore, even though rDNA CN displays very strong familial correlations, it remained possible that blood cell composition somehow influences rDNA CN. Such reverse causality effects are known to be an issue for some multi-copy elements such as mtDNA copy number (mtDNA CN). Neutrophils are known

(B) Effect size (left) and significance level (right) for the association between 18S Ratio and blood cell composition ratios (neutrophil-to-lymphocyte ratio [NLR], platelet-to-lymphocyte ratio [PLR], systemic immune-inflammation index [SII], and lymphocyte-to-monocyte ratio [LMR]) in WB participants in the first WGS release.

(C) Schematic illustration of reverse causality as potential explanation for observed associations between CN and blood cell composition. (i) This schematic represents the issue of reverse causality when measuring mtDNA CN in whole blood (20 hypothetical cells). Mitochondria are generally less abundant in neutrophils (green) compared with other blood subtypes (gray). When the proportion of neutrophils increases (e.g., from 30% to 70%), the measured mtDNA CN is apparently lower. The small vertical lines depict a hypothetical mtDNA CN of 2 in neutrophils and 3 in other cells. (ii) This schematic represents a hypothetical example of what might be observed if reverse causality was influencing the association between rDNA CN and blood cell composition (20 hypothetical cells). Given the positive association between rDNA CN and neutrophil counts, that would entail neutrophils having more copies of rDNA than other cell types. Therefore, if the neutrophil proportion increased (e.g., from 30% to 70%), the measured rDNA CN would also be apparently higher. The small vertical lines depict a hypothetical rDNA CN of 4 in neutrophils and 3 in other cells. (iii) Solely based on the results in (A) and (B), the direction of causality in the association between rDNA CN and blood cell composition could be in either direction (arrow 1, rDNA CN causally influencing blood cell composition; or arrow 2, blood cell composition influencing rDNA CN, such as in (ii); NLR used here for illustration). We then consider a context, such as aging, in which we know NLR changes (arrow 3; note the direction of the association between aging and NLR cannot be in the reverse direction). If reverse causality underlies the association between rDNA CN and NLR, i.e., arrow 2, then we should observe an indirect association between the context under consideration, e.g., aging and rDNA CN, i.e., arrow 4.

(D) Significance levels for the association between known drivers of blood cell composition changes and NLR on all WB participants (light gray) and WB participants with WGS data available in the first sequencing release (dark gray), as well as with 18S Ratio (black). The lack of association between the considered contexts and 18S Ratio despite existing for the associations with NLR suggests reverse causality is not at play here (no arrows 4 and 2 in C(iii) above).

(E and F) (E) Seasonal and (F) circadian patterns on NLR and sequencing center-adjusted 18S Ratio for WB participants. Numbers in brackets represent the number of participants included in each 18S Ratio group. The apparent monthly fluctuations in 18S Ratios are likely random noise (ANOVA p = 0.3).

(G) Mean and corresponding 95% confidence interval of the sequencing center-adjusted 18S Ratio for WB individuals split into five quantiles according to their NLR, PLR, and SII. Increasingly dark shades of red indicate higher values of the corresponding trait.

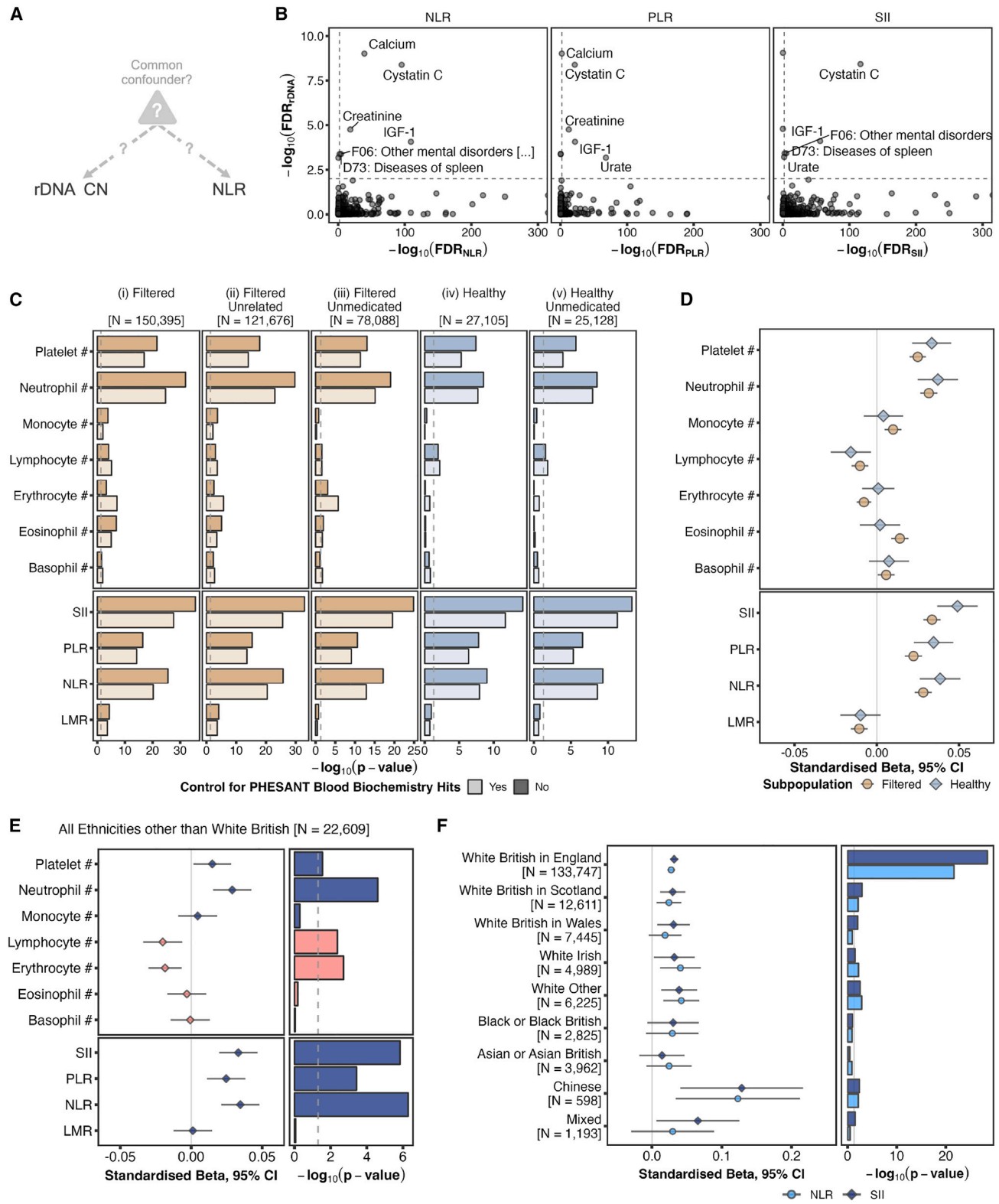

(legend on next page)

**Cell Genomics**
Article

to harbor fewer mitochondria relative to other blood cell subsets.[4,24] Therefore, when measuring mtDNA CN in whole blood, the relative proportion of blood cell subsets, in particular the proportion of neutrophils, has a significant impact on the measured value for mtDNA CN. This is illustrated schematically in Figure 2C(i). If the neutrophil proportion in whole blood increases, as observed in aging and many diseases, then the measured mtDNA CN is lower. Indeed, as demonstrated recently by Gupta et al. using the UKB data, many previously reported trait associations with mtDNA CN disappear after adjustment for blood cell composition.[4]

We therefore considered the possibility that the rDNA CN associations with blood cell subsets may also be due to similar reverse causal effects. The positive association between neutrophil counts and rDNA CN could arise if, hypothetically, neutrophils harbor more copies of rDNA than other blood subtypes (Figure 2C(ii)). Due to the lack of robust associations between rDNA CN and variants elsewhere in the genome, a Mendelian randomization (MR) approach cannot be used to assess causal directionality from rDNA CN to blood cell subtype counts. Given the breadth and size of the UKB dataset, however, we were able to design a range of alternative analyses to test for potential reverse causality.

As noted above, it is well established that neutrophil counts, and NLR, are altered in different contexts including sex, aging, and a range of diseases[25] (we focus on NLR because it is the best studied of the markers). Therefore, if measured rDNA CN is simply a downstream consequence of changes in NLR (i.e., blood cell composition), with neutrophils harboring more copies of rDNA in our hypothetical model, then in any context where NLR increases we should also observe higher rDNA CN (Figure 2C(iii)). However, this does not occur in any of the contexts we considered (Figures 2D and S13). Despite clear associations between the considered traits and NLR (in gray), none of them are associated with rDNA CN (in black). It has also been shown in the UKB that NLR displays circadian and seasonal fluctuations,[26] but again there was no concerted rDNA CN change in either case (Figures 2E and 2F; note, on the other hand, mtDNA CN measured in whole blood does co-vary seasonally, consistent with reverse causal effects[24]). In addition, although MR cannot be used to assess causal directionality from rDNA CN to blood cell subtype counts, an MR analysis can be performed to ask if there is a significant influence of blood cell composition on rDNA CN using previously reported genetic variants for blood cell subtype counts and percentages (from the GWAS Catalog). This analysis found no evidence for

blood cell composition influencing rDNA CN (Table S5; STAR Methods). It is also worth noting that the rDNA CN vs. blood cell subset association appeared monotonic across the entire range of marker values, not just extreme values that might indicate acute disease or severe phenotype (Figure 2G). Therefore, whereas Figures 2A and 2B show that rDNA CN and NLR associate significantly, the results in Figures 2C–2G demonstrate that NLR levels do not influence rDNA CN. The direction of the association thus cannot be from NLR to rDNA CN, ruling out reverse causality as an explanation for the observed association.

We then considered the possibility of a common confounder that simultaneously, but separately, influences both rDNA CN and blood cell subset counts (Figure 3A). First, we observed that technical factors known to affect blood cell counts[27]—delay between venepuncture and count measurements (ANOVA $p = 0.82$) or machine drift over time (ANOVA $p = 0.42$)—were not associated with rDNA CN. We then assessed any potential influence of biological confounders by overlapping phenome-wide screens for rDNA CN with those for NLR, SII or PLR, identifying seven distinct associations that were FDR significant in both cases (Figure 3B; Tables S6–S8). Extending the list of considered associations beyond FDR-significant ones to perform an even more comprehensive analysis of potential confounders revealed all top 15 categorized as either "blood biochemistry," diseases, or medications in the UKB. To assess the influence, if any, of these potential confounders, we filtered the data, first by removing all samples associated with any hematological disorder or that display aberrant blood cell composition measurements ("Filtered" subset derived using the criteria in previous publications[27–29]; Figure 3C(i)), and then, from this, keeping only unrelated individuals (Figure 3C(ii)) and only those not recorded as taking any medication with $p$ value <0.01 in CN associations (Figure 3C(iii)). In all cases, the associations with rDNA CN remained strongly significant. Further sub-setting the dataset to remove all individuals with any recorded disease (at any time prior to recruitment), BMI > 30 kg/m², and smokers, retained significant associations for neutrophils, platelets, lymphocytes, SII, NLR, and PLR ("Healthy" subset; Figures 3C(iv) and 3C(v)). Redoing the analyses using "blood biochemistry" variables as covariates did not diminish the associations. This emphasizes that the rDNA CN influence is on normal variation of blood cell subset counts, and not just a general inflammatory state. To note, although $p$ values naturally weaken as the subsets decrease in size, the effect sizes strengthened further in the "Healthy" subset (Figure 3D).

**Figure 3. Validation of identified associations between rDNA CN and blood cell composition**
(A) Schematic representation of an unknown confounder simultaneously influencing both rDNA CN and a phenotype of interest, such as NLR.
(B) Comparison between the significance levels in phenome-wide screens for blood cell composition ratios (NLR, PLR, and SII) and 18S Ratio in WB participants from the first WGS release. Labeled phenotypes reach FDR < 0.01 in both axes.
(C) Significance levels for associations between 18S Ratio and blood cell composition counts (top) or blood cell composition ratios (bottom) in progressively more stringent WB subpopulations. Light-colored bars indicate that the corresponding linear model includes Calcium, Urate, IGF-1, Cystatin C, Creatinine, and C-Reactive Protein as covariates.
(D) Comparison between effect sizes for the dark-colored (i) "Filtered" and (iv) "Healthy" models in (B).
(E) Effect size (left) and significance level (right) for associations with 18S Ratio in $N = 22,609$ UKB participants from the first WGS release of non-WB self-reported ethnicities.
(F) Effect size (left) and significance level (right) for the 18S Ratio associations with NLR and SII in UKB participants of different self-reported ethnicities, as well as WB participants from different UK countries.

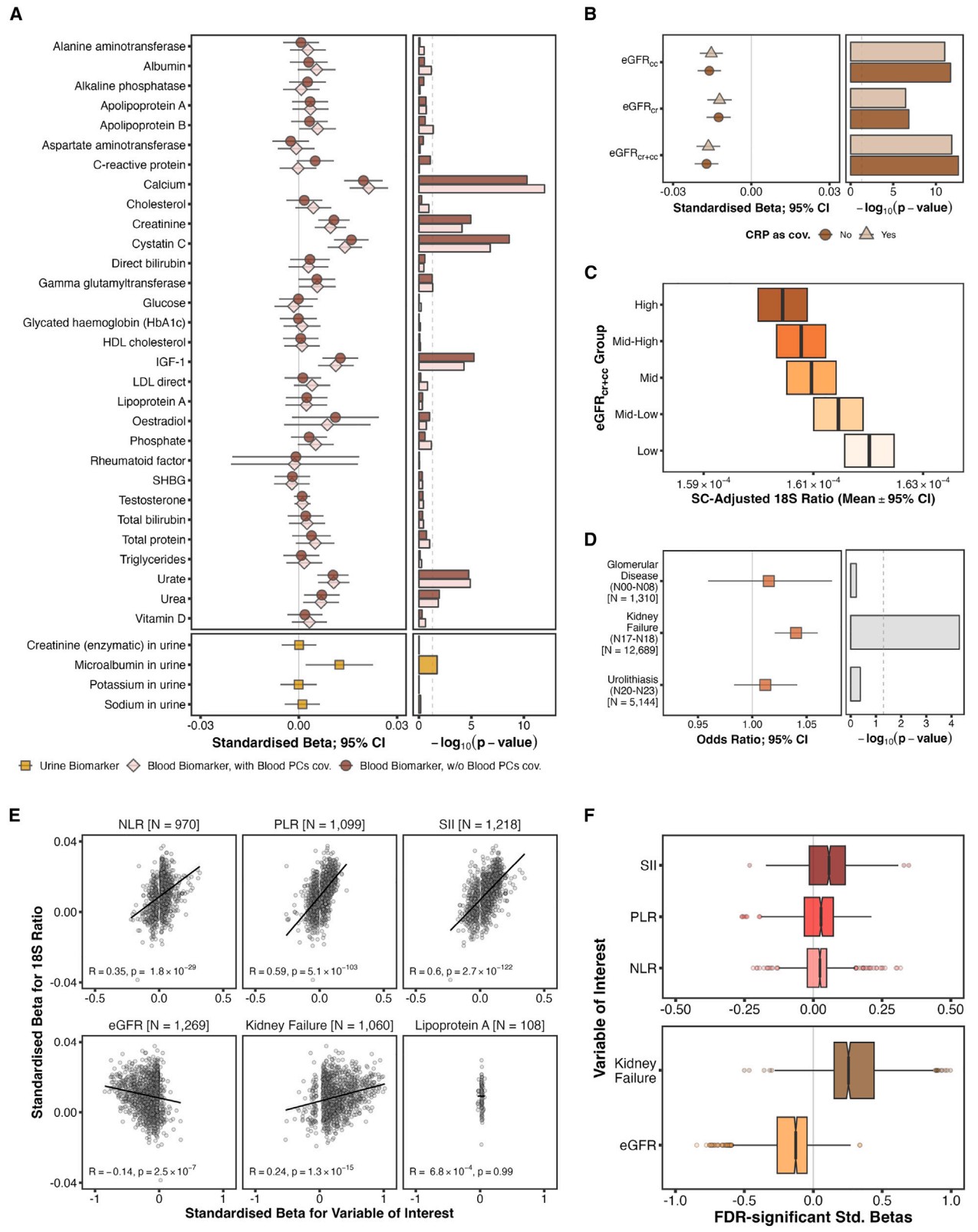

But, could there still be a factor during early development that directionally changes rDNA CN and separately also influences adult phenotypes? Firstly, this would have to be a very common factor as the rDNA CN vs. blood subtype associations spans the entire rDNA CN range (Figure 2G). Second, the very high MZ twin correlations (Pearson's R = 0.95) are relevant here. In the original zygote, prior to the twinning event, there is of course only a single rDNA CN value. So, if rDNA CN genetically changes in a directional manner due to an environmental influence, at any point from the zygotic stage onwards, the highly correlated adult MZ twin rDNA CN values could only arise if the common environmental confounding factor acts either (1) prior to the twinning event, to change rDNA CN genetically, and this is the value inherited by both twins, or (2) after the twinning event, and the rDNA CN simultaneously changes genetically in the two different fetuses. Whatever mechanism is involved in this case, it would have to be extremely finely tuned to genetically change the rDNA CN in the same direction to an almost identical extent in most, if not all, cells in both individuals. In both cases, this factor would also need to influence adult phenotypes separately. We consider both these alternative scenarios to be unlikely and, collectively, our data do not support the existence of a confounder in the association between rDNA CN and blood cell subset counts. Nevertheless, detailed mechanistic studies could be carried out in the future to further probe the issue of potential confounding, if it exists.

### Analysis of rDNA CN and blood cell subset profiles in non-WB ancestries

Finally, we replicated the key rDNA CN vs. blood cell subset findings in 22,609 individuals of non-WB self-reported ethnicity (Figure 3E). Despite the substantially smaller sample numbers, rDNA CN associations with SII and NLR remained statistically significant in most of the White sub-categories, Chinese, and with a clear positive trend in the others (Figure 3F). The differences in the strength of the association among the ethnicities may reflect true biological differences, or be due to, for example, population stratification within each group, smaller sample numbers, or somehow be related to differences in the range of rDNA CNs.

### Association between rDNA CN and renal function

"Blood biochemistry" represented the other major phenotypic category identified in the rDNA CN phenome-wide screen, with five FDR-significant associations (Figure 2A). Since the phenome-wide screen did not account for factors potentially affecting these measurements, we reanalyzed rDNA CN associations with all blood and urine biochemistry biomarkers. For this analysis, we fit linear models on WB individuals not recorded as taking statins, since statins are known to affect several biomarker measurements, including additional covariates—fasting time, extraction date and time, assay date, and sample dilution rate—based on previously published UKB GWASs[30] (Figure 4A; although including individuals on statins yielded virtually identical results for most markers; Figure S14). Given the association with blood cell composition described above, we fit the blood biomarker models with and without blood principal components as covariates. All previous FDR-significant hits remained statistically significant in either case, suggesting that blood cell composition does not confound the results, and both Urea and Microalbumin in Urine reached nominal significance (Figures 4A and S15).

Most putative associations with biochemistry biomarkers related to renal function (Creatinine, Cystatin C, Urea, Urate, and Microalbumin in Urine). Of these, Cystatin C is considered to be the most reliable indicator of renal function.[31] Indeed, Cystatin C remained the most strongly associated with rDNA CN in a multivariate linear model including all blood biochemistry biomarkers related to renal function simultaneously, regardless of whether they appeared as nominally significant on their own (Figure S16A; this occurs both with and without C-Reactive Protein, a well-established marker of inflammation, as a covariate).

Glomerular Filtration Rate values estimated from Cystatin C and/or Creatinine measurements from blood (eGFR) are a widely recognized method for assessing renal function.[30] We thus asked whether rDNA CN associates with eGFR, yielding significantly negative effects regardless of the underlying biomarker, particularly with estimates derived from both (eGFR$_{cr+cc}$; $N$ = 150,378, β = −0.0172 [−0.0217 to −0.0126], $p$ = 2.31 × 10$^{-13}$; Figures 4B and S16B). Similar to Figure 2D, this association appeared monotonic, spanning across the entire range of values (Figure 4C). Since low eGFR values are indicative of kidney failure, we considered the possibility that rDNA CN associates with renal disease. Grouping the renal disease codes described in a previous study[32]—glomerular disease (N00 to N08), kidney failure (N17 and N18), and urolithiasis (N20 to N23)—revealed a significant positive association between rDNA CN and kidney failure ($N_{cases}$ = 12,689, odds ratio [OR] = 1.04 [1.021–1.06],

---

**Figure 4. Association between rDNA copy number and renal function**

(A) Effect size (left) and significance level (right) for the 18S Ratio associations with each blood and urine biochemistry biomarker in the UKB, obtained from linear models with targeted covariates fit on WB individuals from the first WGS release not recorded as taking statins. Results from models for blood-derived biomarkers are reported including and excluding principal components from nucleated blood cell-type proportions as covariates.

(B) Effect size (left) and significance level (right) for 18S Ratio associations with Glomerular Filtration Rate estimates (eGFR) from different methods, both including and excluding C-Reactive Protein levels as covariate.

(C) Mean and corresponding 95% confidence interval of the sequencing center-adjusted 18S Ratio for WB individuals from the first WGS release split into five quantiles according to their eGFR$_{cr+cc}$ value. Increasingly dark shades of orange indicate higher values of the eGFR estimate.

(D) Odds ratio (left) and significance level (right) for the 18S Ratio logistic model associations with groupings of renal diseases in WB UKB participants from the first WGS release.

(E) Pearson's correlation between the effect sizes at FDR < 0.01 obtained on the proteomic associations for each of five traits of interest (NLR, PLR, SII, eGFR$_{cr+cc}$, and Kidney Failure) plus a negative control (Lipoprotein A), and the corresponding effect sizes on the proteomic associations with 18S Ratio on WB individuals. $N$ values indicate the number of proteins from the first UKB OLink release reaching FDR significance for that trait.

(F) Comparison between the proteomic effect sizes for the blood cell composition (top) and renal function (bottom) traits in (E).

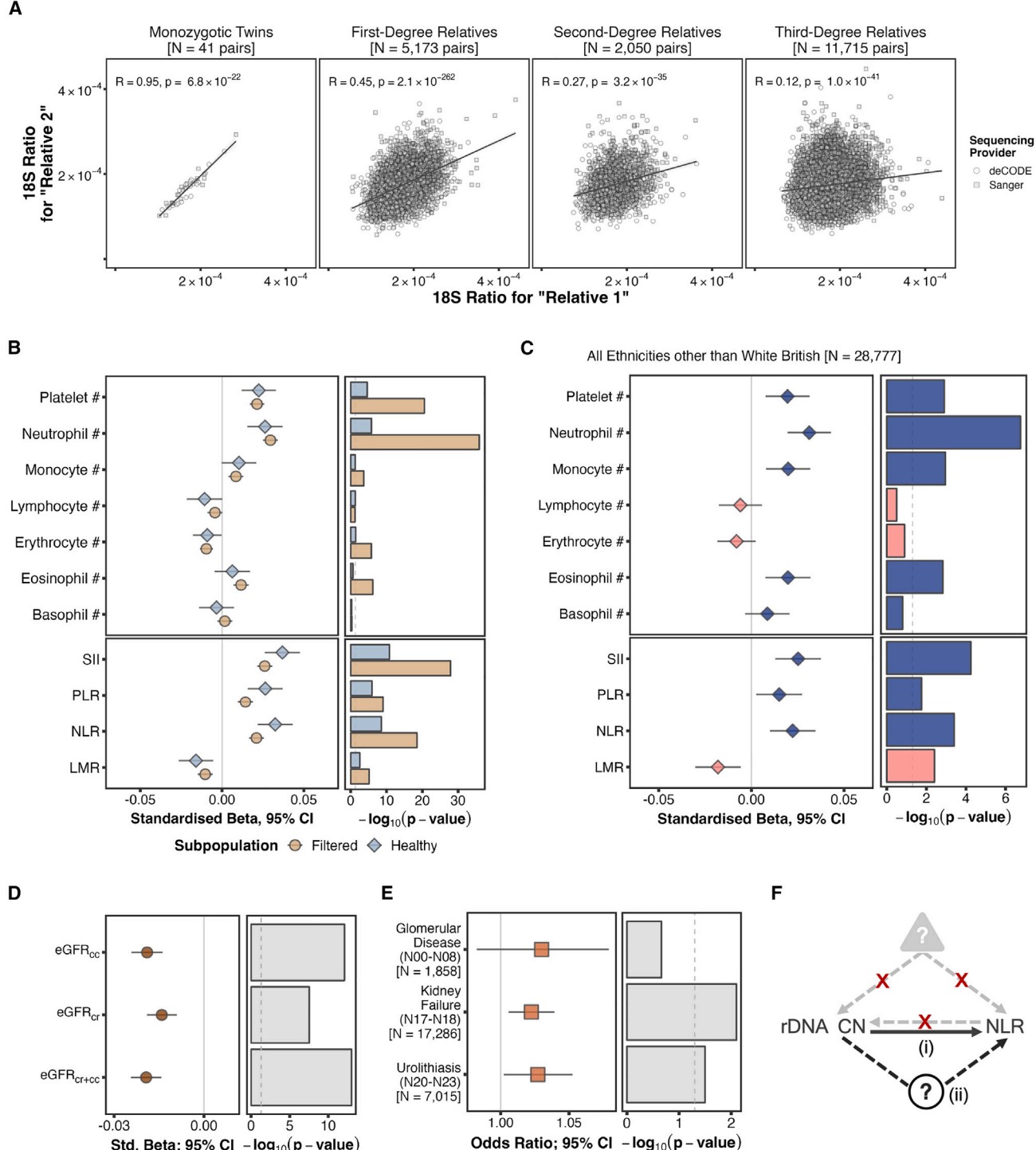

**Figure 5. Replication in the second WGS release**

(A) Correlation between 18S Ratios in genetically identified relative pairs according to KING kinship thresholds[20] sequenced in the same sequencing center batch of the second WGS release.

(B) Effect sizes (left) and significance levels (right) for associations between 18S Ratio and blood cell composition counts (top) or ratios (bottom) in "Filtered" ($N$ = 192,801) and "Healthy" ($N$ = 35,290) WB subpopulations from the second WGS release.

(C) Effect size (left) and significance level (right) for associations with 18S Ratio in $N$ = 28,777 UKB participants from the second WGS release of non-WB self-reported ethnicities.

(D) Effect size (left) and significance level (right) for 18S Ratio associations with eGFR from different methods.

*(legend continued on next page)*

$p = 4.89 \times 10^{-5}$; Figure 4D), suggesting high rDNA CN to be a novel genetic risk factor for renal disease in WB individuals. A similar effect can be observed in participants of non-WB ethnicity, albeit non-statistically significant due to the much smaller sample size ($N_{cases}$ = 1,811, OR = 1.04 [0.987–1.096], $p$ = 0.136).

### Analysis of rDNA CN-associated serum proteomic profiles

Recently, the UKB released serum proteomic profiles for approximately 10% of all individuals in the study.[33–35] Different traits and diseases represented in the UKB are associated with specific serum proteomic profiles.[33–35] If proteomic profiles associated with the hematologic or renal traits are similar to the proteomic profiles associated with rDNA CN, then this would be consistent with the rDNA CN associations we described above being genuine.

The proteomic data were generated using the antibody-based OLink Explore 3072 platform that measures 2,941 protein analytes, representing 2,923 unique proteins. Of these, we considered the 1,463 included in the first release. We used proteomics data for the 35,753 WB individuals available, of which 16,813 were included in the rDNA CN analyses above. We focused on eGFR and kidney failure as renal function traits, and the blood cell composition ratios most strongly associated with rDNA CN—NLR, PLR, and SII—as hematological traits. We also required a "negative control" trait for our analysis. The expectation was that rDNA CN-associated proteomic profiles would correlate with the proteomic profiles associated with the hematological and renal traits discussed above, but not with the proteomic profiles for a trait that is not associated with rDNA CN at the genetic level. However, the negative control trait also had to fulfill several other essential criteria. First, both blood subtype and blood biochemistry profiles can be altered in many different physiological contexts. Any trait that influences these profiles would not be appropriate as a control since we would end up observing apparent correlations between rDNA CN-associated proteomic profile and the negative control trait proteomic profile simply because of downstream consequences (since we find rDNA CN to be an influence on blood subtype and blood biochemistry profiles). Second, the majority of traits in the UKB correspond with very low $N$ values. Using these as controls would be inappropriate since there would be insufficient power to find any significant hits in the associated proteomics profiles in the first place, given that proteomics data are available for only ~10% of the overall dataset. We therefore restricted ourselves to traits associated with at least 100,000 individuals included in the previous rDNA CN analyses. Third, many traits in the UKB do not seem to alter serum proteomic profiles. These include continuous traits that do not show significant quantitative association with serum proteomic profiles, or binary/categorical traits that are not associated with proteomic differences among the different categories. These

would not be useful as control. We therefore chose "Lipoprotein A" as a control that fulfills the above criteria.

For each of the six traits, we then fit linear models for all 1,463 proteins and compared the effect size for their FDR-significant hits (Tables S9–S14) against those obtained in models fit with rDNA CN (Table S15). Figure 4E shows that the effect sizes for NLR, PLR, SII, kidney failure, and eGFR are significantly correlated with the proteomic profile associated with rDNA CN. Notably, all correlations with traits previously shown to associate with rDNA match the expected orientation, with eGFR and kidney failure reversing signs, consistent with low eGFR values reflecting worse renal function (Figure S17). In addition, these analyses reveal that, although most proteomic effects are positive for both blood ratios and kidney failure (and, conversely, negative for eGFR), the magnitudes of the effects are larger for the renal function-related phenotypes (Figure 4F). In contrast, the proteomics profile of the control variable shows no correlation with the rDNA CN profile. Collectively, the above analysis provides an orthogonal validation of the association between rDNA CN and the hematological and renal traits presented above.

It is important to note that the serum proteomic profile is influenced by different types of cells and biological processes, and does not just reflect active protein synthesis. Even in the case of protein synthesis, we cannot distinguish potential indirect effects, e.g., if an mRNA for a repressor is translated at a higher efficiency and that then causes lower levels of transcription, and hence ultimately lower protein levels, from downstream target genes (which may also explain the presence of some negative effects). Establishing the precise contribution of rDNA CN to the proteomic profiles associated with kidney function and/or blood cell composition will require further detailed mechanistic investigations beyond the scope of this study.

### Replication in second UKB WGS release

In December 2023, the UKB released WGS data for an additional ~300,000 individuals of whom 209,681 are of WB ancestry and fulfill the same criteria employed for the first release. We processed these sequences in a similar manner to the original samples and used them to replicate the key observations. In the second release, highly significant differences in mean 18S ratio estimates remained between sequences generated by deCODE and Sanger (Figure S18; note, there is no "Vanguard" batch in this release). Nevertheless, comparison of unadjusted 18S ratios in twin pairs suggested harmonization between sequencing centers had improved. In particular, the Pearson's R for pairs with both twins sequenced in the second release (Pearson's R = 0.92) is higher than when both twins were sequenced in the first release or one in each (Pearson's R = 0.83 in both cases; Figure S19). When considering relative pairs in which both individuals were sequenced in the same center, results from the second release are virtually identical to those from the first one (from Pearson's R= 0.95 for twins to R = 0.12 for third-degree relatives; Figure 5A).

---

(E) Odds ratio (left) and significance level (right) for the 18S Ratio logistic model associations with groupings of renal diseases in WB UKB participants from the second WGS release.

(F) Schematic representation of the potential (i) direct or (ii) indirect causal influences of rDNA CN on human phenotypes, such as NLR. The red crosses indicate that our data do not support the involvement of either reverse causality or confounding effects.

A GWAS analysis of rDNA CN on 162,283 unrelated WB individuals from the second WGS release yielded two whole-genome-significant variants in chromosome 15 (15:20861938_ CCAAT_C with beta = $6.61 \times 10^{-5}$, SE = $1.02 \times 10^{-5}$, $p$ = $1.1 \times 10^{-10}$; and rs113939809 with beta = $6.16 \times 10^{-5}$, SE = $1.1 \times 10^{-5}$, $p$ = $2.1 \times 10^{-8}$; Figure S20; GWAS Catalog accession GCST90356220). In both cases, however, their effects in the unrelated GWAS for the first release did not even reach nominal significance (beta = $1.06 \times 10^{-5}$, SE = $1.2 \times 10^{-5}$, $p$ = 0.38; and beta = $1.02 \times 10^{-5}$, SE = $1.32 \times 10^{-5}$, $p$ = 0.44, respectively). Moreover, rs62153030, the single whole-genome-significant variant in the first release (when related individuals were included in the analysis) also failed to reach nominal significance in the second release (beta = $-2.32 \times 10^{-7}$, SE = $4.64 \times 10^{-7}$, $p$ = 0.62). The inconsistency of these results further suggests the spurious nature of the putative hits, reinforcing the idea that rDNA CN variation is not driven by variation elsewhere in the genome.

The key associations between rDNA CN and blood cell composition measurements observed in the first release are replicated in the second release. As Figure 5B shows, neutrophil and platelet counts, as well as NLR, PLR, and SII remain highly significantly associated with rDNA CN in a consistent direction for both "Filtered" (N = 192,801) and "Healthy" (N = 35,290) WB subpopulations generated as above. Neutrophil counts are the top hit among the blood cell count measurements in both cases (β = 0.0296 [0.025–0.034], $p$ = $1.31 \times 10^{-36}$ for "Filtered"; β = 0.0263 [0.016–0.037], $p$ = $1.65 \times 10^{-6}$ for "Healthy"), with SII being the most strongly associated among the blood cell composition ratios (β = 0.0261 [0.0215–0.0307], $p$ = $1.34 \times 10^{-28}$ for "Filtered"; β = 0.0369 [0.0262–0.0476], $p$ = $1.60 \times 10^{-11}$ for healthy). Similarly consistent results are also obtained among non-WB participants (N = 28,777; Figure 5C).

In the case of renal function associations, consistent with the results obtained in the first sequencing release, eGFR values derived from Creatinine and/or Cystatin C measurements negatively associate with rDNA CN, with eGFR$_{cr+cc}$ yielding the strongest association (N = 199,588, β = $-0.0193$ [$-0.0244$ to $-0.0142$], $p$ = $9.06 \times 10^{-14}$; Figure 5D). This is again accompanied with an increased risk of kidney disease (Figure 5E), in this case for both urolithiasis (N$_{cases}$ = 7,015, OR = 1.03 [1.00–1.05], $p$ = 0.0318) and kidney failure (N$_{cases}$ = 17,286, OR = 1.02 [1.01–1.04], $p$ = 0.00797).

## DISCUSSION

Our analyses support rDNA CN variation being a genetic influence on hematological profiles and renal function in humans. At this stage, further mechanistic investigation is limited by the lack of suitable methods for the controlled genetic manipulation of rDNA CN in mammalian cells, but there are several, not necessarily mutually exclusive, speculative mechanisms by which rDNA CN could influence cellular outcomes. First, there could be a variety of effects on translational outcomes, including protein levels as suggested by the analysis of the UKB proteomics data (pathway (i) in Figure 5F). An alternative mechanism could be more indirect (pathway (ii) in Figure 5F), whereby rDNA CN acts as a "sink" for epigenetic modifier proteins, indirectly influencing gene expression in the rest of the genome.[36] In support of this, rDNA CN has

been shown to associate with steady-state gene expression differences of single-copy genes in human cell lines.[5] It is also important to consider that tissue-specific influences of rDNA CN could occur via interaction with tissue-specific expression of ribosomal biogenesis factors and/or ribosomal proteins. For example, many ribosomopathies—diseases caused by genetic mutations in ribosomal constituents or in factors with a role in ribosome assembly—are associated with tissue-specific effects, often including hematopoietic defects (reviewed by Kampen et al.[37]). Furthermore, Murre and colleagues found that, during human neutrophil differentiation, rDNA is progressively sequestered at the lamina in a heterochromatic environment and mature neutrophils do not synthesize rRNA. They proposed that the life span of mature neutrophils may be influenced in part by variation in residual rRNA abundance.[38] It is possible that rRNA abundance is influenced by rDNA CN. These are all speculative examples by which rDNA CN might exert its effects, and it is quite possible that other completely independent, and as yet unknown, mechanisms are involved. In the future, it will be important to perform large-scale analyses that provide a more direct measure of cellular translation in the cell type of interest using methods such as Ribo-Seq.[39]

The impact of rDNA CN on phenotypes is likely an underestimate for several reasons. First, rDNA units are not all genetically identical to each other. They harbor both SNVs and INDELs whose prevalence might lead to distinct phenotypic outcomes, and thus associations at more granular levels than the total number of copies will need to be explored. Second, we did not consider epigenetic states. In the mouse, it is known that specific rDNA haplotypes are associated with increased DNA methylation levels,[19] thereby potentially altering the "effective" CN in terms of what is actually expressed. Third, the rDNA CN proxy estimates undoubtedly retained some measurement error, thus potentially reducing the strength of the associations that could be detected. Fourth, phenome-wide screens have far greater statistical power for identifying associations with continuous measurements. Fifth, the UKB is not representative of the wider UK population, since participants are relatively older and live in more socioeconomically advantaged areas, with a sampling bias for "healthier," white, and female individuals.[20] Finally, discovery analyses were conducted solely on the first WGS release, limiting the statistical power to identify rDNA-associated traits. Future studies could leverage the entire UKB WGS dataset for discovery, along with other WGS datasets, for even greater power to find additional associations.

In summary, we report a novel source of genetic variation influencing human phenotypes. In the absence of profiling rDNA, the proportion of trait variance that should be attributed to rDNA-associated genetic variation could mistakenly be considered as "intangible variation." Rather, it is genetic variation that has thus far been overlooked, acting in addition to genetic variation in the rest of the genome to influence phenotypic outcomes in humans.

### Limitations of the study

There are two key limitations to our work: (1) although we find no evidence for reverse causality or common confounders influencing our results, there could still be common factors influencing both rDNA CN and phenotype but are not detectable in the UKB data, and (2) we have performed a comprehensive

## Article

**CellPress**

analysis of only a single ancestral group (WB) and, although we present evidence of replication in other ancestries, in the future this needs to be done in other large non-UKB WGS datasets.

## STAR★METHODS

Detailed methods are provided in the online version of this paper and include the following:

- KEY RESOURCES TABLE
- RESOURCE AVAILABILITY
  - Lead contact
  - Materials availability
  - Data and code availability
- EXPERIMENTAL MODEL AND SUBJECT DETAILS
- METHOD DETAILS
  - Variables of interest and covariates
  - Development and validation of 18S Ratio as rDNA CN proxy estimate
  - Total rDNA CN equivalents
  - Sequencing center adjustment
  - Ethnicity-specific subsets
  - Relatives
  - Subsets derived from the White British population
  - Genome-wide associations
  - Phenome-wide screens
  - Mendelian randomisation
  - Extraction and analysis of OLink proteomics data
- QUANTIFICATION AND STATISTICAL ANALYSIS
  - General statistical analysis
  - Regression models

## SUPPLEMENTAL INFORMATION

## ACKNOWLEDGMENTS

We acknowledge the assistance of the ITS Research team at Queen Mary University of London. This research utilized QMUL's Apocrita HPC facility, supported by QMUL Research-IT (https://doi.org/10.5281/zenodo.438045). We thank Prof. David van Heel and Dr. Michelle Holland for feedback on the manuscript. F.R.-A. and V.K.R. were supported by grants from the Biotechnology and Biological Sciences Research Council (BB/R00675X/1), Barts Charity (MGU0390 and G-002588), and Rosetrees Trust (100182). D.M.E. was supported by a National Health and Medical Research Council (NHMRC) Investigator Grant (APP2017942). This research was conducted using the UK Biobank Resource (application 83271).

## AUTHOR CONTRIBUTIONS

Conceptualization, data curation, and formal analysis was performed by F.R.-A. and V.K.R. Software was developed by F.R.-A. Methodology was developed by F.R.-A., D.M.E., and V.K.R. Funding was acquired by D.M.E. and V.K.R. Overall coordination was provided by V.K.R. The manuscript was written by F.R.-A., D.M.E., and V.K.R.

## DECLARATION OF INTERESTS

The authors declare no competing interests.

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

## STAR★METHODS

### KEY RESOURCES TABLE

| REAGENT or RESOURCE | SOURCE | IDENTIFIER |
|---|---|---|
| **Deposited data** | | |
| 1000 Genomes rDNA CN estimates | Hall et al.[16] | Supplementary Information 1 |
| 18S Ratio estimates for UK Biobank participants | This paper | Return for UKB application 83271 |
| 18S Ratio GWAS summary statistics | This paper | GWAS Catalog: GCST90356215, GCST90356216, GCST90356217, GCST90356218, GCST90356219, GCST90356220 |
| Lymphocyte percentage of white cells GWAS summary statistics | Astle et al.[27] | GWAS Catalog: GCST004632 |
| Lymphocyte percentage of white cells GWAS summary statistics | Vuckovic et al.[28] | GWAS Catalog: GCST90002389 |
| Neutrophil count GWAS summary statistics | Astle et al.[27] | GWAS Catalog: GCST004629 |
| Neutrophil count GWAS summary statistics | Keller et al.[40] | GWAS Catalog: GCST002557 |
| Neutrophil count GWAS summary statistics | Chen et al.[29] | GWAS Catalog: GCST90002351 and GCST90002355 |
| Neutrophil count GWAS summary statistics | Kachuri et al.[41] | GWAS Catalog: GCST90056178 |
| Neutrophil count GWAS summary statistics | Barton et al.[42] | GWAS Catalog: GCST90025977 |
| Neutrophil count GWAS summary statistics | Vuckovic et al.[28] | GWAS Catalog: GCST90002398 |
| Neutrophil count GWAS summary statistics | Sakaue et al.[43] | GWAS Catalog: CST90018968 |
| Neutrophil percentage of white cells GWAS summary statistics | Astle et al.[27] | GWAS Catalog: GCST004633 |
| Neutrophil percentage of white cells GWAS summary statistics | Vuckovic et al.[28] | GWAS Catalog: GCST90002399 |
| Platelet count GWAS summary statistics | Astle et al.[27] | GWAS Catalog: GCST004603 |
| Platelet count GWAS summary statistics | Guo et al.[44] | GWAS Catalog: GCST010229 |
| Platelet count GWAS summary statistics | Auer et al.[45] | GWAS Catalog: GCST009465 |
| Platelet count GWAS summary statistics | Read et al.[46] | GWAS Catalog: GCST008168 |
| Platelet count GWAS summary statistics | Ferreira et al.[47] | GWAS Catalog: GCST000510 |
| Platelet count GWAS summary statistics | Li et al.[48] | GWAS Catalog: GCST001783 |
| Platelet count GWAS summary statistics | Gieger et al.[49] | GWAS Catalog: GCST001337 |
| Platelet count GWAS summary statistics | Shameer et al.[50] | GWAS Catalog: GCST002186 |
| Platelet count GWAS summary statistics | Chen et al.[29] | GWAS Catalog: GCST90002357 and GCST90002361 |
| Platelet count GWAS summary statistics | Hu et al.[51] | GWAS Catalog: GCST90026525 |
| Platelet count GWAS summary statistics | Barton et al.[42] | GWAS Catalog: GCST90025951 |
| Platelet count GWAS summary statistics | Little et al.[52] | GWAS Catalog: GCST90094627 |
| Platelet count GWAS summary statistics | Kachuri et al.[41] | GWAS Catalog: GCST90056183 |
| Platelet count GWAS summary statistics | Vuckovic et al.[28] | GWAS Catalog: GCST90002402 |
| Platelet count GWAS summary statistics | Sakaue et al.[43] | GWAS Catalog: GCST90018969 |
| **Software and algorithms** | | |
| TrimGalore | Krueger[53] | https://github.com/FelixKrueger/TrimGalore |
| bowtie2 | Langmeade and Salzberg[54] | https://github.com/BenLangmead/bowtie2 |
| samtools | Li et al.[55] | https://github.com/samtools/samtools |
| BWA-MEM | Li[56] | https://github.com/lh3/bwa |
| BOLT | Loh et al.[57] | https://alkesgroup.broadinstitute.org/BOLT-LMM |

*(Continued on next page)*

*Continued*

| REAGENT or RESOURCE | SOURCE | IDENTIFIER |
|---|---|---|
| plink2 | Chang et al.[58] | https://www.cog-genomics.org/plink/2.0/ |
| PHESANT | Millard et al.[59] | https://github.com/MRCIEU/PHESANT |
| TwoSampleMR | Hemanni et al.[60] | https://mrcieu.github.io/TwoSampleMR/ |
| Analysis scripts used in this study | This paper | Zenodo: https://doi.org/10.5281/zenodo.10938487 |

## RESOURCE AVAILABILITY

### Lead contact
Further information and requests should be directed to and will be fulfilled by the lead contact, Vardhman Rakyan (v.rakyan@qmul.ac.uk).

### Materials availability
This study did not generate new unique reagents.

### Data and code availability
Individual-level 18S Ratio estimates generated for this study have been returned to the UK Biobank, and summary statistics for 18S Ratio GWASs have been deposited in the GWAS Catalog (https://www.ebi.ac.uk/gwas/).
   Analysis scripts developed for this study are publicly available at https://doi.org/10.5281/zenodo.10938487.

## EXPERIMENTAL MODEL AND SUBJECT DETAILS

The present study associates phenotypic, genetic, and proteomic data from over 500,000 UK Biobank participants recruited between 2006 and 2010, obtained via project application 83271. The subpopulations created for each particular analysis are described below in the appropriate Method Details subsections.

## METHOD DETAILS

### Variables of interest and covariates
Although some variables are only of interest in specific analyses and will be described more in detail in their respective sections, those that appear commonly as covariates and/or traits of interest are described here.

   Sex was obtained from UKB field ID 31, with a value of "0" representing females and "1" males. Age at recruitment was obtained from the first instance of field ID 21003, from which Age Squared was also derived as the value multiplied by itself. The Assessment Center is recorded in field ID 54, and the Adjusted Telomere Length in field ID 22191. From the Genetic Principal Components in field ID 22009, only the first 10 are used as covariates in all analyses. Increasing to all 40 available principal components provides virtually the same results, even in the presence of high genetic heterogeneity (Figure S21). In that case, heterogeneity across ethnicities is largely controlled by including self-reported Ethnic Background from field ID 34 as additional covariate.

   Sequencing Center is an essential covariate to control when considering rDNA CN estimates from the UKB. For each sample in the first WGS release, the sequencing center batch was obtained from the RG field in the alignment file header. In particular, samples with the string "CN:SC" included within RG were assigned to the "Sanger", with those without "Main Phase" in the same field further assigned to the "Sanger Vanguard" batch; otherwise, samples were assigned to "deCODE". Both "Sanger" and "Sanger Vanguard" were combined into a single category of the Sequencing Center variable. For the second WGS release, Sequencing Center information was obtained from the information available in field ID 32051.

   In GWAS analyses of rDNA CN, additional covariates intended to control for potential differences in blood cell composition were also included. This was achieved by obtaining principal components from the percentages of Lymphocytes (field ID 30180), Monocytes (field ID 30190), Neutrophils (field ID 30200), Eosinophils (field ID 30210), and Basophils (field ID 30220). All individuals with more than 0% in Nucleated Red Blood cells (field ID 30230) were discarded from the analysis, leaving 472,269 participants. These were then used as input for the PCA() function from the FactoMineR package[61] version 2.8 with parameters scale.unit = TRUE, graph = FALSE, and ncp = 10, providing 5 dimensions for each individual. Of these, the first four explain over 99.99% of the variance and were thus included as covariates in the GWASs as well as in associations with blood biochemistry markers.

Cell counts were obtained for Platelets (field ID 30080), Erythrocytes (field ID 30010), Lymphocytes (field ID 30120), Monocytes (field ID 30130), Neutrophils (field ID 30140), Eosinophils (field ID 30150), and Basophils (field ID 30160). Blood cell composition ratios were derived as follows:

$$NLR = \frac{Neutrophil\ counts\ (30140)}{Lymphocyte\ counts\ (30120)}$$

$$PLR = \frac{Platelet\ counts\ (30080)}{Lymphocyte\ counts\ (30120)}$$

$$LMR = \frac{Lymphocyte\ counts\ (30120)}{Monocyte\ counts\ (30130)}$$

$$SII = \frac{Neutrophil\ counts\ (30140) \times Platelet\ counts\ (30080)}{Lymphocyte\ counts\ (30120)}$$

To avoid deriving infinity values that are unsuitable for the linear modeling functions employed, these values were not calculated for individuals with 0s recorded for Lymphocytes or Monocytes.

Temporal variables related with acquisition and processing of blood samples were considered in some analyses. Time and date for venepuncture and processing were retrieved from field ID 3166 and 30142, respectively. The delay between them was calculated as their difference. Month and hour of venepuncture used for seasonal and circadian patterns were parsed from field ID 3166.

Blood and urine biomarker measurements were obtained from the fields in category IDs 17518 ("Blood biochemistry") and 100083 ("Urine assays"), respectively, whereas their assay dates and sample dilutions were obtained from category IDs 18518 ("Blood biochemistry processing") and 148 ("Urine processing"). In addition, urine sampling date was retrieved from field ID 20035, and fasting time from field ID 74.

GFR estimates derived from Creatinine and/or Cystatin C measurements were calculated using the most recently-accepted formulas, according to the National Kidney Foundation (https://www.kidney.org/professionals/kdoqi/gfr_calculator). In particular,

$$eGFR_{cr} = \gamma_{cr} \times min\left(\frac{S_{cr}}{\kappa}, 1\right)^{\alpha_{cr}} \times max\left(\frac{S_{cr}}{\kappa}, 1\right)^{-1.2} \times 0.9938^{Age}$$

$$eGFR_{cc} = \gamma_{cc} \times min\left(\frac{S_{cc}}{0.8}, 1\right)^{-0.499} \times max\left(\frac{S_{cr}}{0.8}, 1\right)^{-1.328} \times 0.996^{Age}$$

$$eGFR_{cr+cc} = \gamma_{cr+cc} \times min\left(\frac{S_{cr}}{\kappa}, 1\right)^{\alpha_{cr+cc}} \times max\left(\frac{S_{cr}}{\kappa}, 1\right)^{-0.544} \times min\left(\frac{S_{cc}}{0.8}, 1\right)^{-0.323} \times max\left(\frac{S_{cc}}{0.8}, 1\right)^{-0.778} \times 0.9961^{Age}$$

where $S_{cr}$ corresponds to the serum Creatinine measurement in field ID 30700 multiplied by 0.0133 to convert its units from μmol/L to mg/dL, as required by the formula; $S_{cc}$ is the Cystatin C measurement in field ID 30720 (already in mg/dL); Age is the age of recruitment from field ID 21003; $\gamma_{cr}$ is 142 for males and 143.704 for females; $\gamma_{cc}$ is 135 for males and 130.035 for females; $\gamma_{cr+cc}$ is 133 for males and 123.956 for females; $\alpha_{cr}$ is 0.9 for males and 0.7 for females; and $\kappa$ is 0.9 for males and 0.7 for females.

Dates for the first reports of renal disease were obtained from fields in category ID 2414 ("Genitorurinary system disorders"). The sole entry with a recorded date ("1903-03-03") among those with special meanings according to UKB data coding 819 was discarded. The remaining dates were then discretised as representing "cases" for their given disease code, with empty entries indicating "controls". Only those participants considered as "controls" for all disease codes within a given combined disease group (i.e., no valid date in any of the corresponding field IDs) were also deemed as "controls" for the resulting group.

### Development and validation of 18S Ratio as rDNA CN proxy estimate

Previously-published rDNA CN estimates from sequencing data rely on alignments to tailored genome assemblies to identify rDNA reads, and to filtered reference exomes to provide a baseline for single-copy read depth.[5,19] Given the unfeasibility of obtaining such alignments for all UKB sequences, and being aware of published studies having relied on rDNA pseudocopies and unplaced scaffoldings existing in human assemblies to extract rDNA reads from previously-aligned data,[17] we developed a proxy estimation method suitable for UKB alignments. To this end, we leveraged the available samples from the 1000 Genomes Project British in England and Scotland (GBR) population as a testbed. Figure S1 depicts the procedure schematically.

We obtained paired-end WGS fastq files stored at the corresponding sequence_read folder within the gridftp/1000g/ftp/phase3/data/ path for the 94 GBR individuals available through the EMBL-EBI public endpoint in Globus, except those generated using ABI SOLiD technologies. (13 individuals listed as having low-coverage WGS data available in the 1000 Genomes Project website,

https://www.internationalgenome.org/data-portal/population/GBR, were not present on September 9th, 2022, when the data was retrieved, and remain unavailable at the moment of writing). These files were then merged according to their sample and strand of origin, and trimmed using trimgalore[53] version 0.6.5 in paired-end mode with all other parameters as default. The trimmed sequences were then aligned in two different ways to either obtain rDNA CN estimates according to previously validated methods or reproduce the alignment conditions in the UKB.

First, trimmed reads were aligned with bowtie2[54] version 2.4.5 to the same reference sequences and following the same procedure as described by Rodriguez-Algarra et al.[19] In brief, reads were first aligned to a tailored Hg38 assembly. In this, all regions identified as similar to the rDNA were masked. Moreover, an entire rDNA unit reference (obtained from GenBank accession KY962518.1) was included as additional contig after being tweaked to reduce potential loss of coverage around the Transcriptional Start Site (TSS). To this end, the rDNA sequence spanning from a breakpoint 2120 base pairs upstream of the rDNA TSS (corresponding with the midpoint of the repetitive element closest to the 3′ end of the unit) and up to the end of the original reference was prepended to the sequence from the TSS up to the breakpoint. This so-called "looped" rDNA reference then covers a single entire unit, but with the start and end of the sequence moved upstream from the promoter. Trimmed reads were also aligned to the human exome reference (labeled 9606 on the EMBL/EBI online repository) with sequences from sex chromosomes, smaller than 300 base pairs, and those deemed as significantly similar by blastn[62] version 2.7.1+ in --ungapped mode removed. The average read depth per sample was calculated on these alignments from the output of samtools depth[55] version 1.10. rDNA CN was then obtained as twice the quotient between average read depth at the 18S subunit from the tailored whole genome alignments and the value per sample obtained from the exome alignments.

Second, to mimic the UKB alignments, trimmed reads were also aligned with BWA-MEM[56] version 0.7.17 to the unmodified Hg38 assembly (GRCh38.p13). Prior to alignment, however, singleton reads, which remain in the data despite their mate having been previously discarded due to insufficient post-trimming length, were removed using the repair.sh script from BBMap[63] version 38.95, with the -Xmx20g options. BWA-MEM was then executed with parameters -K 100000000 -Y, and its output piped to samtools sort to obtain bam files sorted by position, which were then indexed with samtools index. These alignments were then used to extract the number of rDNA reads in each sample as well as the number of reads mapped to numbered chromosomes, which served as baseline for the rDNA CN proxy estimates. For the former, the exact location of rDNA analogues in Hg38 is necessary. In particular, the region corresponding to the 18S subunit was extracted from the KY962518.1 reference sequence (positions 3658 to 5526) and blasted against Hg38's chromosome 21, as well as the chr22_KI270733v1_random and chUn_GL000220v1 scaffolds. These three contigs harbor in the BWA-MEM alignments the vast majority of reads that bowtie2 assigns to the rDNA (Figure S22). The regions that blastn reported as 18S analogues (4 on chromosome 21, and 2 on each of the other two scaffolds; Table S16) were then saved as a bed file and used as input for a samtools view -c call with options -M, -L <bedfile>, and -X <bamfile> <index> to count the total number of 18S reads. These values were then divided by the sum of reads assigned to numbered chromosomes as reported by samtools idxstats to obtain the 18S Ratio used as rDNA CN proxy. Note that different versions of Hg38 might contain a different number of scaffolds, so further (or fewer) 18S analogues might be present in other datasets.

On the UKB, 18S Ratios were obtained as described above for all WGS samples available in CRAM format for both first and second releases. To this end, alignment files were accessed through the UK Biobank Research Analysis Platform (UKB RAP). These are stored in subfolders within the /Bulk/Whole genome sequences/Whole genome CRAM files/ path for the first WGS release, and /Bulk/GATK and GraphTyper WGS/Whole genome GATK CRAM files and indices [500k release]/ for the second. To enable parallelisation of the computation, the paths to the CRAM files were split into batches and saved in separate files with up to 10,000 participants each. The read counts were then retrieved using samtools through the UKB RAP application called swiss-army-knife (SAK) version 4.5.0. In particular, two scripts were implemented, one to be executed locally and another stored within the UKB RAP and called from the former. The communication between the two scripts was done through the dxpy python library installed on a virtual environment using pip install dxpy. This provides the dx command line tool, which can then be used from within the local script to launch multiple SAK instances in the UKB RAP after successful login. These SAK instances were executed including the options -iin="${project}${script_path}" and -icmd="bash '${script}' '$i'", where ${project} is the result of dx pwd, i.e., the working directory in the UKB RAP, ${script_path} is the location of the remote script within the UKB RAP workspace, ${script} is the file name of the remote script, without the entire path, and $i is the index of the particular instance to be executed. The path to a folder where the results should be saved was also provided using the --destination option. The remote script then reads the list of CRAM paths corresponding to its instance number and, for each, calls samtools to count the number of reads in the region of interest. To avoid the CRAM files needing to be copied, which would seriously hamper the efficiency of the computation, their paths were prepended with/mnt/project/. The resulting output files including the counts were then downloaded and processed locally to calculate the 18S Ratio for each participant.

### Total rDNA CN equivalents

Equivalent total rDNA CN values were derived from 18S Ratios calculated on all UKB WGS samples included in the first WGS release with self-reported White British ethnic background ("1001" in the first instance of field ID 21000) and Caucasian background confirmed through genotyping ("1" in field ID 22006). To this end, calculated 18S Ratios were then multiplied by the total sum of the numbered chromosome sizes in Hg38 (2,875,001,522 bases) and divided by the size of the 18S subunit (1,871 bases). The

distribution of these resulting values was then compared with the rDNA CN estimates derived from 18S that Hall et al.[16] report for 85 of the GBR individuals from the 1000 Genomes Project.

### Sequencing center adjustment

Sequencing-center adjustment of 18S Ratios was necessary for some analyses. After joining the two "Sanger" sequencing batches, 18S Ratios were adjusted by subtracting from "Sanger" estimates the magnitude of their corresponding regression coefficient from a linear model fit with all samples from the first WGS release including only sequencing center as independent variable. This removes the statistical significance in the difference of their means (Figure S2). The main text refers to these values as "rDNA CN estimates adjusted for sequencing center", whereas several figure axes present them as "SC-adjusted 18S Ratio".

### Ethnicity-specific subsets

All 502,384 UKB participants were assigned to ethnic supergroups according to their self-reported ethnicity, as recorded in the first instance of field ID 21000. In particular, values of "1", "1001", "1002", and "1003" were assigned to "White"; values of "2", "2001", "2002", "2003", and "2004" were assigned to "Mixed"; values of "3", "3001", "3002", "3003", and "3004" were assigned to "Asian or Asian British", values of "4", "4001", "4002", and "4003" were assigned to "Black or Black British"; and values of "5" were assigned to "Chinese". Every other participant was considered belonging to "Other ethnic group". All 199,779 participants from the first WGS release with 18S Ratio estimates were included in analyses of ethnicity-specific rDNA CN differences.

Analyses explicitly mentioning "White British" individuals only retain those that have both "1001" in field ID 21000 and "1" in field ID 22006 (166,919 individuals in the first WGS release). All analyses involving principal components from genotypes reduce this subpopulation further to remove those whose genotyping array was listed as "BiLEVE" (negative values in field ID 22000). The final set was comprised of 157,227 WB individuals from the first WGS release (and 209,681 on the second release replication). Regression analyses involving other ethnicities also remove "BiLEVE" array samples. WB individuals were split by UK country of birth according to the value in field ID 1647 ("1" for "England", "2" for "Scotland", and "3" for "Wales"; other values in this field were discarded). "White Irish" and "White Other" individuals correspond to values "1002" and "1003" in field ID 21000, respectively.

### Relatives

At the moment of writing, UKB provides genetic relatedness information for 107,076 pairs of participants, with 147,612 individual participants represented. Pairs with kinship >0.4 were considered monozygotic twins, between 0.177 and 0.354 first-degree relatives, between 0.0844 and 0.177 second-degree relatives, and between 0.0442 and 0.0884 third-degree relatives, according to the reported KING estimates.[20,64] Comparison between 18S Ratios in relatives were limited to pairs where both individuals belonged to the "White British" subset described above.

First-degree relatives were further classified into "Fraternal" and "Parental" relationships. In particular, pairs with IBS0 value lesser than 0.0012 were deemed "Parental", and the rest were considered as "Fraternal". Out of caution, five pairs with IBS0 lesser than 0.0012 – hence theoretically "Parental" relationships – were discarded since their births were less than 10 years apart (according to the values recorded in field ID 34).

Each individual on the relatedness table was also assigned to a "family" following the process Figure S23 exemplifies, separating first and second WGS releases. First, the older individual of each pair was identified by their birth year (when birth year coincided, the individual marked as number 2 on the relatedness table was arbitrarily assigned as "older" for this purpose). Then, ids for all "younger" individuals paired with each "older" relative were grouped alongside the id for the corresponding "older" individual. Although this step is not mandatory, since each relationship pair could be considered its own group, it substantially increases the efficiency of the computation on the 107,076 pairs. The groups were then used to construct a sparse matrix with the sparseMatrix() function of the Matrix package[65] version 1.5.3, on which a cross-product was calculated with the tcrossprod() function of the same package. The result of the cross-product was converted into a graph with the graph_from_adjacency_matrix() function from the igraph package[66] version 1.4.1, and then employed as input for the clusters() function of the same package. The membership field of the clusters() output provided a family ID for each of the relationship groups, which could finally be used to assign each individual participant to a family by reconstructing who belonged to which group. These families were then employed to identify a subset of $N = 127,231$ completely unrelated WB individuals in the first WGS release, and $N = 162,682$ in the second, by keeping only those without entry in the relatedness table and the oldest ones from each family group.

### Subsets derived from the White British population

The WB individuals used for association analyses were further subset to more stringent criteria. The subpopulations we denote "Filtered" recapitulate the exclusion criteria followed in previously-published blood-composition association analyses in the UK Biobank.[27–29] To this end, participants were excluded if they had any of the blood-related cancer codes listed in Table S17 recorded in the first instance of field ID 20001 and/or any of the non-cancer blood-related diseases listed in Table S18 recorded in the first instance of field ID 20002. Moreover, all individuals pregnant at the moment of recruitment (or with pregnancy status unknown; values "1" and "2" of field ID 3140) were also excluded, as well as those with potentially-aberrant blood-related measurements. In particular, participants were required to have a maximum of $200 \times 10^9$ leukocyte cells per blood liter (field ID 30000), 20 g/dL of hemoglobin (field

ID 30020), 60% of hematocrit (field ID 30030), and $1,000 \times 10^9$ platelets per blood liter (field ID 30080). Finally, all participants for which their blood samples were processed more than 36 h after venepuncture were also discarded.

For the first WGS release participants, the "Filtered" subpopulation was further subset in two different ways. In particular, the "Filtered Unrelated" subpopulation only included participants not listed in the genetic relatedness table or who were the oldest from each identified family. The "Filtered Unmedicated" subpopulation, on the other hand, excluded participants with any of the medication codes listed in Table S19 recorded in the first instance of field ID 20003 and/or "taking other prescription medications" (a "1" in the first instance of field ID 2492).

The "Healthy" subpopulations excluded from the "Filtered" individuals any participant with any entry in the first instance of field ID 20001 (cancer) or 20002 (non-cancer disease). Furthermore, all participants with any entry recorded in the first instance of any field belonging to category ID 2417 ("Congenital disruptions and chromosomal abnormalities", listed in Table S20) were also excluded. Finally, only participants listed as not being smokers at the time of recruitment (values "0" or "1" in field ID 20116) and with a maximum BMI of 30 kg/m$^2$ (field ID 21000) were retained in the "Healthy" subpopulation. For the first WGS release, the same criteria were then used to generate the "Healthy Unmedicated" subpopulation from the "Filtered Unmedicated" one described above.

Participants recorded as taking any of the statin medications listed by Sinnott-Armstrong et al.[30] – Simvastatin (1140861958), Fluvastatin (1140888594), Pravastatin (1140888648), Lipitor 10mg tablet (1141146138), Atorvastatin (1141146234), or Rosuvastatin (1141192410) – on the first instance of field ID 20003 were removed for the associations between rDNA CN and blood biochemistry in Figures 4A and S15. 24,575 participants were removed for that reason. The equivalent analysis in Figure S14, on the other hand, includes all 157,227 WB individuals from the first WGS release.

### Genome-wide associations

GWASs were conducted with BOLT-LMM[57] version 2.4 on a set of variants pre-processed with plink[58] version 2.0-20220602. In particular, files for chromosomes 1 to 22 from the "22822 UKB imputation from genotype" release were initially filtered separately using plink2 with parameters --autosome, --max-alleles 2, --mac 20, --geno 0.05, --hwe 1e-6, and --maf 0.001 (which, as mentioned below, was increased to 0.01 after merging) for each WGS release separately on all participants with alignment data available. For reference, on the first WGS release, this reduced the number of chromosome 1 variants from 7,402,791 to 1,086,793, with 5,850 variants removed due to missing genotype data, 439,555 removed due to the Hardy-Weinberg exact test threshold, and 5,800,593 removed due to the minimum allele frequency threshold. Despite including the --max-alleles option, some multi-allelic variants appeared to be retained, causing errors in further steps. For that reason, all variants with multiple entries in the resulting pvar file were tagged and explicitly removed using plink2 --exclude. On chromosome 1, this removed a further 1,445 variants in the first WGS release, leaving a total of 13,763,355 variants across chromosomes.

Chromosome-specific files were then merged in plink2 with options --mind 0.1 and --maf 0.01, leaving 7,234,608 variants available for association analyses in the first WGS release. Prior to executing BOLT-LMM on the WB individuals, however, a subset of independent variants was identified with plink2 --indep-pairwise 200 50 0.25 and its output linked in the --modelSnps option of bolt --lmm. This enables the model to be fit solely on those independent variants, but associations to be assessed on the entire variant set. Linkage information was provided through the --LDscoresFile option with LDSCORE.1000G_EUR.tab.gz (included in the BOLT-LMM release). Sex, assessment center, and sequencing center were included as qualitative covariates, and age, age squared, the first 10 genetic principal components, the first 4 blood composition principal components, and adjusted telomere length were included as quantitative covariates, alongside associated options --covarUseMissingIndic and --covarMaxLevels 1000. The genomic inflation factor of the resulting associations was calculated in R using the formula:

$$\lambda_m = \frac{\text{median}(\text{qchisq}(1 - \text{pval}, 1))}{\text{qchisq}(0.5, 1)}$$

where qchisq(p, df) calculates the quantile function of a $\chi^2$ distribution with df degrees of freedom at probability values p, and pval is the vector of significance values recorded at the P_BOLT_LMM_INF column of the BOLT-LMM output.

BOLT-REML was executed exactly as reported for BOLT-LMM above on the subset of completely unrelated WB individuals, replacing --lmm with --reml. The SNP heritability coefficient $h_g^2$ and corresponding standard error was retrieved from the Variance component 1: "modelSnps" entry in the output log file.

### Phenome-wide screens

Phenome-wide screens were conducted using PHESANT,[59] a software toolbox implemented in R and specifically designed for that purpose in the UKB, on the set of 157,227 WB participants from the first WGS release, as well as on the subsets of individuals whose WGS data was sequenced in either deCODE or the Sanger. Fields belonging to the 67 UKB categories listed in Table S21 were considered for their inclusion. For some categories, however, only some fields ended being included in the analysis. In particular, all fields with any of the strings "record", "Reason", "Source", "Interpolated", "Method of recording", or "(pilot)" in their titles were discarded. From category ID 100094 ("Baseline characteristics"), only the "Townsend deprivation index at recruitment" was retained. Note this particular field currently appears in UKB's search engine listed with ID 22189, but both our phenotype table and PHESANT's variable list refer to it as ID 189. From category ID 100011 ("Blood pressure"), only fields with "automated" in the title were retained (field IDs 102, 4080, and 4081). From category ID 100033 ("Early life factors"), field IDs 120, 1647, 1767,

and 20115 were discarded. Finally, from category ID 100043 ("Hearing"), only field ID 2247 ("Hearing difficulty/problems") was retained.

The selected fields were then processed to enable their use as phenotypes in PHESANT, which only considers values recorded in the first instance (i.e., entries in the phenotype table of the form <field ID>-0.X), so every field without first-instance entries was discarded. Table S22 lists the final set of 1,280 field IDs employed in the analyses. Moreover, variable names were transformed as requested, prepending an "x" and replacing dashes with underscores.

The phenomeScan.r script from PHESANT was executed for each reported trait of interest (18S Ratio, NLR, PLR, and SII) using the exact same input and settings, but phenotypes from the "Blood count" category (ID 100081) were only considered for downstream analyses of 18S Ratio associations. Sex, age, age squared, sequencing center, assessment center, the first 10 genetic principal components, and adjusted telomere length were used as covariates, and the --genetic="FALSE" option was included, since the principal components were explicitly provided in the covariate table. The covariate table was subset accordingly for the sequencing centre-specific analyses of 18S Ratio associations, and its rows shuffled for the positive control analysis in Table S2, whereby each entry of the phenotype table did not match the corresponding 18S Ratio estimate and corresponding covariates. The script was run for each analysis in 250 parts, which were then combined using the mainCombineResults.r script. Among the provided fields, PHESANT explicitly excluded those from category ID 265 ("Telomeres") due to them being "genetic". Field IDs 20002 ("Non-cancer illness code, self-reported"), 40001, and 40002 ("primary" and "secondary" causes of death, respectively) were also excluded due to them having been "superseded" by more specific variables, and field ID 87 was excluded because of it being "polymorphic". Overall, PHESANT generated output for 2,722 distinct phenotypes. To note, the total number of phenotypes exceeds that of input fields since specific ones, such as the medications in 20003, lead to multiple "phenotypes".

From the combined results, entries of varType "CAT-SIN" that included a "-" character in their varName were discarded, since PHESANT is not able to provide effect size estimates for these (all reported as −999). The number of cases in categorical variables was extracted from the n column, as well as the total number of participants with reported values for each entry. Only phenotypes with at least 200 cases and value recorded for at least half of the total number of individuals were considered for the analyses presented in the main figures, including the FDR computation – a total of 1,078 phenotypes. Moreover, for presentation purposes, some variable names were shortened and/or edited to remove typos. For the sake of completeness, however, Tables S1–S4 and S6–S8 report the entire unaltered results as provided by PHESANT.

### Mendelian randomisation

Previously-reported variants associated with blood composition were retrieved from the GWAS Catalog[67] (file gwas_catalog_v1.0.2-associations_e108_r2023-01-14.tsv) to construct "exposure" sets for Mendelian Randomisation (MR). Entries from this table corresponding with phenotypes of interest were identified according to the MAPPED_TRAIT and DISEASE/TRAIT columns. To best match the population of interest, only those studies that included the strings "European" or "British" in the INITIAL SAMPLE SIZE variable were retained, and all entries with the string "NR" in the same variable were discarded. (Studies employed for each phenotype are shown in the key resources table). At the time of writing, no suitable associations were identified for leukocyte counts or systemic inflammation markers. In particular, entries for NLR associations did not include effect sizes or standard deviations.

Standard deviations derived from Beta and p-values were calculated for each association using the get_se() function from the TwoSampleMR package[60] version 0.5.6, as are all other functions mentioned in this section. The value stored in the column STRONGEST SNP-RISK ALLELE was split into SNP and effect_allele, as required by the format_data() function. When multiple associations were reported for the same combination of SNP and effect_allele, only the one with minimum p-value and/or maximum Beta was retained. These were further reduced after applying clump_data() to retain only variants not in linkage disequilibrium.

The BOLT-LMM output for the GWAS from the first WGS release WB participants (deposited in GWAS Catalog accession **GCST90356215**) was used to construct the "output" data required for the MR analysis. Similar to the exposure data, this was also transformed using the format_data() function matching column names with parameters: snp_col = "SNP", beta_col = "BETA", se_col = "SE", eaf_col = "A1FREQ", effect_allele_col = "ALLELE1", other_allele_col = "ALLELE0", and pval_col = "P_BOLT_LMM_INF". Exposure and outcome data were then joined using the harmonise_data() function, which further discards "ambiguous" variants, either because of "incompatible alleles" and/or "being palindromic with intermediate allele frequencies". Finally, MR itself was conducted using the mr() function with the method = "mr_ivw" option to employ the Inverse Variance Weighted procedure.

### Extraction and analysis of OLink proteomics data

Normalised Protein eXpresion (NPX) values for the first release of UK Biobank's OLink proteomics data were retrieved as follows. First, on the UKB RAP, subsets of individuals were selected using the Cohort Browser on the latest available data release, using "Number of proteins measured | Instance 0 > 0" and "Ethnic Background | Instance 0 IS ANY IF 'British'" as filtering criteria. In the case of 18S Ratio associations, "Coverage (from QC metrics for WGS processing) > 90" was also included as filtering criterion. These subsets were then saved and later employed on the Table Exporter tool within the RAP, alongside a text file including protein names obtained from DNA Nexus' github page (https://github.com/dnanexus/UKB_RAP/blob/main/proteomics/field_names.txt) on the 12th of August, 2023. To generate a table of proteomics NPX values, the Table Exporter also required explicitly including the value "olink_instance_0" in the Entity field within the Advanced Options. The tables resulting from submitting these jobs were then saved to

be analyzed in R. The overall cohort included 46,020 participants when generated on the 8[th] of November, 2023 after the second releases of both proteomics and sequencing data became available. Of these, 35,754 had values recorded for the variables of interest, the control and its covariates and were used for the analysis. In the case of 18S Ratio associations, 16,813 participants from the first WGS release were included.

Linear regression models were then fit to determine the potential association between each NPX value as response variable and either 18S Ratio, one of the five variables of interest (NLR, PLR, SII, eGFR, and Kidney Failure) or the control (Lipoprotein A) as explanatory variable, including sex, age, age squared, assessment center, and 10 first genetic PCs as covariates in both cases, plus sequencing center and adjusted telomere length for 18S Ratio. In addition, the proteomics Plate ID (UKB field ID 30901) of each individual was also included as a further categorical covariate in either case. Effect sizes for the associations with the variables of interest or the control at FDR-adjusted significance level <0.01 were then compared with their corresponding effect sizes in the 18S Ratio associations using Pearson's correlation, as presented in Figure 4E.

## QUANTIFICATION AND STATISTICAL ANALYSIS

### General statistical analysis

Unless otherwise specified, all analyses were conducted using in-house scripts implemented in the R programming language version 4.3.1, with data manipulation and visualisation packages included in tidyverse[68] version 2.0.0. Difference of means was assessed using Wilcoxon signed-rank tests from R's wilcox.test() function, including the paired = TRUE option when suitable. Correlation coefficients and *p*-values were derived using Pearson's method from the cor.test() function, and ANOVA *p*-values were obtained using the Anova() function from the car package[69] version 3.1.2 over lm() linear regression models. ANOVA significance levels for continuous variables exactly match that of the underlying linear model coefficient. Beta estimates and corresponding *p*-values for these coefficients were calculated on models fit with all continuous-valued variables (both dependent and independent) normalised using R's scale() function. Odds ratios were calculated as $e^{\beta}$ using R's exp() function from effect size estimates obtained on logistic regression models fit using R's glm() function with the parameter family = "binomial". Details for the regression models can be found later in the "Regression models" subsection. Confidence intervals for the effect sizes were calculated using the tidy() function from the broom package version 1.0.5, with parameters conf.int = TRUE and conf.level = 0.95. All other confidence intervals were calculated using the formula $\overline{X} \pm t_{((1-\alpha/2),N-1)} \times (\hat{\sigma}_X / \sqrt{N})$, where $\overline{X}$ indicates the mean of the variable of interest, $\hat{\sigma}_X$ its empirical standard deviation, $N$ the sample size, and $t_{((1-\alpha/2),N-1)}$ the $1 - \alpha/2$ quantile of a Student's t distribution with $N - 1$ degrees of freedom, $\alpha$ being 0.05 for a 95% interval, calculated with the qt() function. Quantiles for particular variables, such as the blood cell composition ratios and eGFR values being divided into Low, Mid-Low, Mid, Mid-High, and High groups, were obtained with the cut_number() function from the ggplot2 package, specifying n = 5 in this case.

### Regression models

Regression models were fit and analyzed throughout the present study as reported above. This section describes the details specific for each of those models. Unless otherwise specified, linear models with 18S Ratio as explanatory variable were employed. The exceptions were the linear models fit for the associations displayed in Figures S11 and S16A, denoted as "multivariate" in the text. In these, 18S Ratio was included as response, with the variables for which association results are presented were all jointly included as explanatory variables. No assumption of causal directionality is implied in any of these models. They address the extent to which the combination of the variables of interest is able to predict the value of the response, and each individual coefficient relates to the individual contribution of each variable of interest to the overall prediction. In addition, associations between rDNA CN and renal disease groups in Figures 4D and 5E were obtained on logistic regression models with rDNA CN as explanatory variable, and including solely a standard set of covariates consisting of sex, age, age squared, sequencing center, assessment center and adjusted telomere length.

Significance of self-reported ethnic background differences in 18S Ratios was obtained with an ANOVA test over a linear model with the same covariates listed above for the logistic model. Associations with sex and age on WB individuals were analyzed similarly over a single linear model without self-reported ethnic background as covariate but with the first 10 genetic principal components instead.

Single-ethnicity models for the association between 18S Ratio and blood cell composition included sex, age, age squared, sequencing center, assessment center, the first 10 genetic principal components, and adjusted telomere length as covariates. For the multi-ethnic analysis of Figures 3E, 5C, and S21, self-reported ethnic background was included as well.

For Figures 2D and S13, the same variables as above were employed as covariates in the models assessing influences on NLR, but with age and age squared being deemed response variables in their own models. The trait of interest for "Cancer" associations indicates whether an individual has any entry in the first instance of field ID 20001.

In order to assess the potential effects of technical factors related with blood acquisition and processing on 18S Ratios, the linear model employed included the same usual covariates listed above (sex, age, age squared, sequencing center, assessment center, 10 first principal components, and adjusted telomere length) as well as both the delay in processing the blood samples and the machine drift over time. The drift was represented for each individual as the number of days between when their blood sample was processed and the first date registered overall, after having discarded all individuals with over 36 h delay between their venepuncture and blood

processing. In addition, models controlling for potential "Blood biochemistry" confounders included Calcium (field ID 30680), Urate (field ID 30880), IGF-1 (field ID 30770), Cystatin C (field ID 30720), Creatinine (field ID 30700), and C-Reactive Protein (field ID 30710).

For the associations between 18S Ratio and biochemistry biomarkers in Figures 4A, S14, and S15, some variables aside from the standard covariates above were included in the model. In particular, fasting time (field ID 74) and the sample dilution corresponding to each biomarker were incorporated as covariates alongside variables derived from the extraction and assay dates. These were generated as follows to mimic the covariates employed by Sinnott-Armstrong et al.[30] Year, month, hour, minute, and second were parsed from blood (field ID 3166) and urine (field ID 20035) extraction dates. From this, a time of extraction covariate was derived by summing the hour plus the minute divided by 60 and the second divided by 3600. A discretised month of extraction covariate was then derived by combining the month and year values parsed before, except for all dates within 2006 and starting from the first of August 2010, which were included in their own separate categories. The dates of processing, on the other hand, were treated as continuous variables indicating their delay from the earliest recorded for the biomarker. Blood biomarker associations were also calculated with and without including blood-derived PCs (in Figures 4A, S14, and S15) and with and without C-Reactive Protein (in Figures 4B and S16) as covariates.

