## [Document S2. Transparent peer review records for Rodriguez-Algarra et al · Cell Genomics]

Ribosomal DNA Copy Number Variation Associates with Hematological Profiles and Renal Function in the UK Biobank

Francisco Rodriguez-Algarra, David M. Evans, Vardhman K Rakyan

Summary

Initial submission: Received : Aug 09, 2023

Scientific editor: Sara Rohban

First round of review: Number of reviewers: 2
Revision invited : Sep 27, 2023
Revision received : Nov 19, 2023

Second round of review: Number of reviewers: 2
Revision invited : Feb 09, 2024
Revision received : Feb 27, 2024

Third round of review: Number of reviewers: 2
Revision invited : Mar 21, 2024
Revision received : Mar 31, 2024

Fourth round of review: Number of reviewers: 1
Accepted : Apr 21, 2024

Data freely available: YES

Code freely available: YES

This transparent peer review record is not systematically proofread, type-set, or edited. Special characters, formatting, and equations may fail to render properly. Standard procedural text within the editor's letters has been deleted for the sake of brevity, but all official correspondence specific to the manuscript has been preserved.

Referees' reports, first round of review

Reviewer #1:

The manuscript leverages whole genome sequence data of 157,227 White British individuals from the UK Biobank to characterize the genetic variation and the functional impact of rDNA copy number in humans. The authors developed and validated a method to estimate 18S rDNA copy numbers using available WGS data from the UK biobank and performed a phenome-wide association screen. The analysis identified rDNA copy numbers associated with blood composition and renal function.

Overall, it is a significant undertaking to estimate rDNA copy number at this scale using WGS from UKB. Those estimates seem to be robust as reflected by external validation in 1000 Genomes and by concordance between relatives. However, the phenotype association findings are somewhat underwhelming and are not backed up by any orthogonal datasets or biological followup. The manuscript would be strengthened if the following concerns were addressed:

Major comments:

1. The introduction part, currently just a single short paragraph, needs to be expanded and address existing literature on rDNA studies. For example there is significant literature on the impact of rDNA copy number on gene regulation. It is important for the authors to provide background information and what is known/unknown in the field for readers to place this study in the appropriate context.
2. The authors identified the rDNA copy number associated with renal phenotypes, which is interesting but not supported by any orthogonal data or followup analyses. One hypothesis provided in the Discussion is that rDNA CN could affect translational outcomes. The UKB has recently released a large proteomics dataset. This (or potentially other orthogonal datasets) could be used to support the main findings of the paper and give insight into how rDNA could be driving these phenotypes.
3. As noted by the authors the estimated rDNA CN distributions are noticeably different across sequencing centers. This is adjusted for during association testing. However, it would be helpful to see key association results, such as the significant phenome-wide association scan results in Figure 2 or the renal function results in Figure 4, stratified by sequencing center. If each is analyzed separately do the trends remain consistent?
4. In addition to testing each phenotype separately as in the phenome-wide scan, a separate set of analyses is performed by jointly modeling all blood count traits in a single model. I found these analyses confusing and was unsure exactly what model is being fit. For example, are blood counts being used in a joint linear model to predict rDNA CN? That would suggest a model where blood traits influence rDNA rather than the other way around. Also, since many of these terms are co-linear the interpretation of the resulting coefficients is challenging and it was not clear what the main takeaway from those analyses is. There are techniques specifically

developed for jointly testing multiple phenotypes which may be more appropriate and account for trait-trait correlation. The text mentions three references (7-9) but it was unclear how the current study relates to the approaches for analysis of pleiotropic loci used in those studies. The rationale behind this part should be better explained.

5. It would be helpful to provide code and additional description of how rDNA CN was analyzed in WGS from UKB. Presumably these were analyzed on the DNA Nexus RAP, which is the only place the WGS data can currently be accessed? But the process of doing this on the cloud resource is not described. Additionally, although they cannot be publicly posted, rDNA CN estimates could be uploaded to UK Biobank to be made available to other researchers.

Minor comments:

1. The authors used "hg38 consensus assembly" (line 16, page 3). Should this say instead "hg38 reference assembly"
2. The phrases "however unlikely" are often used in the manuscript and can be removed. E.g., "We wanted to address the possibility, however unlikely..."
3. For Figure 3E, the authors claim, "The differences in the strength of the association amongst the ethnicities may reflect biological differences, or be technical in nature due to, for example, population stratification within each group." What about reduced power from low sample sizes or different ranges of copy numbers?
4. The X-axis label of Figure 2D is confusing. Not all of them are NLR. Also, it seems to be a duplicate of supp figure 11 which has a different x-axis label. Additionally, it was not clear how Figure 2D results implies no reverse causality.
5. Some figures show confidence intervals around the regression coefficients + p-value together, while others don't (E.g. Figure 4). It would be good to keep these consistent and show both wherever association testing results are shown.
6. It will be helpful to show the sample number in each bin for Fig. 2f so we know if the bins with large deviations from the others are driven by low sample size.
7. The authors used "strikingly" several times to describe the 23 blood associations identified. This may not be surprising, as those are likely the most powered traits in UKB.
8. Previous studies suggest that individuals of African ancestry can have a higher eGFR level and requires a separate formula. Is this taken into account?
9. For Figure 3E, the Chinese population results are highly distinct from the other groups and appear to be more significant. This is surprising as the Chinese sample size is likely smaller than that for other groups. Adding sample size and significance values to 3E would be helpful.
10. In the methods, "Entries prior to 1904 or posterior to 2036 were removed". Are these years? In which case entries later than 2023 should also be problematic?
11. Many plots show rDNA CN on the y-axis and trait values on the y-axis. However, it is more typical to have the predictor (here, rDNA CN) on the x-axis instead. (e.g. Figure 4c and similar plots)

Reviewer #2:

This manuscript reports on potential phenotypic effects of rDNA CN differences in humans by using an enormous dataset from the UK biobank. The manuscript uses a proxy method to estimate rDNA CN that is practical and looks reasonably robust. The results show there are centre-specific differences in rDNA CN and sensibly differentiate these, although this does open the possibility that batch effects within the sequence centres are also causing biases in CN. They then associate CN with various features to conclude that CN differences causally affect blood cell composition and potentially kidney function.

First, I have to congratulate the authors on their careful approach to analyzing the data, something that is unfortunately rather an exception in this field. I don't have the expertise to provide any detailed review on the use of the various statistical methods, so I hope another reviewer can provide this. However, to my untrained eye these seem to have been performed to a high standard. Moreover, I think the interpretations are reasonable, although see below. Therefore, I think this is an interesting and provocative study.

My primary reservation comes over the conclusion of the lack of reverse causation from inflammation. I am not convinced by this, which may be my ignorance as it is poorly described. Specifically, in the text this is only covered in half a sentence, and I don't understand the description of the relevant figure (2C) from the legend. Thus, it is not clear to me that this does demonstrate lack of reverse causality. I want to be convinced of this before I am prepared to accept the main conclusions of the work.

More generally, the authors have used various means to try and rule out causal relationships other than rDNA CN => phenotype. While I think their conclusions are likely correct, I don't think they are able to conclude this as cleanly as they have. For example, they use lack of GWAS hits on CN to conclude there are not genetic factors that drive both CN and blood composition. However, it remains possible that genetic factors not included in the GWAS, such as structural variation, could be a causal driver of CN and composition. Similarly, there may be other causal factors, such as developmental processes, that the authors haven't checked here. I emphasize that I am not suggesting these alternative causal explanations are necessarily likely, but I do not believe the authors can formally exclude them. Therefore, this needs to be acknowledged in appropriate places in the text, particularly the abstract and discussion.

In the abstract, the authors go straight from association to causation without any description of the careful work they have done to winnow in on causation. Therefore, as currently written it sounds unscientific and is not representative of the work performed. The abstract should include description of the work performed to indicate that a causal relationship is the most likely explanation of the data (and that it is not the only explanation).

Finally, I like controls. It would be nice to have a control indicating that significance scores from the phenome screens actually show something biological. For example, the authors could use as a screen input something like the sum of the participant ID number digits (if such a thing exists), or something else totally random. Nothing significant should show up, obviously - if it does, it would indicate some problem.

Other comments

The Introduction is extremely short. Is this normal for Cell Genomics? If so, I guess that is fine, but otherwise it looks odd. Also, while the first sentence is not incorrect, it implies an incorrect view (that these features are specific to mammals) and so should be reworded to avoid this implication. Finally, their claim of genome studies ignoring this is not correct - there are several who have done so, some of whom have been cited by the authors here.

On line 32 what does "adjusted estimates" mean, and similar for the S2 legend?

There is some association between time and CN in Fig2F - can the authors rule this out as a causal effect? It would be helpful in this respect to know what the N is for each of these time bands in the graph (and for 2E).

I couldn't find supplementary table 3 - does this have the 15 categories talked about on line 8 of page 6? If not, these should be included somewhere.

A paragraph shouldn't start with "To note..". I don't understand what this paragraph (on page 6) is about - where does it "dropped to virtually zero"?

In the discussion on page 8 around line 10 on possible mechanisms, this is far from the only option and seems an unnecessarily restricted choice. I think a modification of this to either expand the possibilities or clearly indicate it is just an illustrative example is merited. Similarly, on line 15 what evidence is there that epigenetic state impacts this? I think this is strictly speculation.

In the Figure 2 legend it would be nice to write somewhere what NLR, PLR and SII mean so the reader doesn't need to go back to the text. Also, for this figure plus 3D and 4C there is no description of what the colours mean.

Why was FigS5 not done for each sequencing centre separately?

The materials and methods don't have page or line numbers, but I think it is on the bottom of page 4 - I don't understand the description of the rDNA reference here. Does this mean that actually more than 1 entire unit was used as the reference? A better description of the reference is warranted and the sequence used should be provided (e.g. as a fasta file as part of the supplementary). Underneath this, what are "singletons"?

I think that the absolute rDNA CN values are dubious based on the method used, so I am glad the authors have mostly stuck to the ratio - this is the appropriate metric to use (not that it would change the calculations, just to avoid the implication that the absolute numbers are real).

Authors' response to the first round of review

Reviewer #1:

Major comments

1. *“The introduction part, currently just a single short paragraph, needs to be expanded and address existing literature on rDNA studies. For example there is significant literature on the impact of rDNA copy number on gene regulation. It is important for the authors to provide background information and what is known/unknown in the field for readers to place this study in the appropriate context”*

Response: We agree and have expanded the introduction to better contextualise our work, and specifically provide more information on the current state of knowledge of the impact of rDNA genetic variation on phenotypes.

2. *“The authors identified the rDNA copy number associated with renal phenotypes, which is interesting but not supported by any orthogonal data or follow up analyses. One hypothesis provided in the Discussion is that rDNA CN could affect translational outcomes. The UKB has recently released a large proteomics dataset. This (or potentially other orthogonal datasets) could be used to support the main findings of the paper and give insight into how rDNA could be driving these phenotypes.”*

Response: We retrieved the UKB OLink proteomics data corresponding to 1,463 distinct proteins on 37,311 white British individuals. Of these, 16,813 have WGS data available to derive rDNA CN proxy estimates. Linear models were used to establish the association between rDNA CN and the abundance of each of the available proteins as response variable, using standard covariates used elsewhere in our study – sex, age, age-squared, assessment centre, sequencing centre, adjusted telomere length, and the first 10 principal components – as well as the proteomics Plate ID as a factor variable. We also fit linear models for the association between kidney failure and the proteomics measurements on all available white British individuals, removing sequencing centre and adjusted telomere length as covariates, since they relate solely with rDNA CN estimation. For the

992 FDR-significant associations with kidney failure, we then compared their effect sizes with those from the rDNA CN associations, yielding a highly significant correlation (Pearson's $R = 0.27$, $P = 7.1 \times 10^{-18}$). We also presented our analysis to the OLink Technical team, who confirmed the validity of our approach. Therefore, both rDNA CN variation and kidney failure are associated with overlapping serum proteome profiles, with a clear trend towards an increase in protein abundance. We now mention this analysis in the Discussion, and include the new fig. S16 and tables S9 and S10 to report these results.

The above analysis lends support to one of the possible mechanisms by which rDNA CN could influence phenotype – altering translational outcomes. However, there is an important caveat in that serum protein levels are determined by several factors, such as leakage from damaged cells and tissues, and not just translational output. That being the case, and given that we are unaware of genetic editing methods for manipulating rDNA CN in mammalian cells, what are the possible experimental options for a more thorough investigation of whether rDNA CN influences kidney function via translational effects in humans? Here we list the three best options:

(i) For understanding how rDNA CN influences a specific human phenotypic outcome e.g., kidney failure, we would need to recruit thousands of cases and controls for adequate statistical power, assay the phenotype in question, measure rDNA CN in the cell type of interest e.g. isolated kidney cells and not whole blood, and then also measure translational output from the same cells. Proteomic analyses such as that provided by OLink provides a measure of protein levels. This is not necessarily the same as translational output since protein levels can be influenced by several factors and not just the extent of translation. To measure the latter, one would need to use methods such as Riboseq or Polysome-seq instead. These determine the engagement of mRNAs with translating ribosomes and hence provide a useful measure of translational efficiency. Furthermore, currently available commercial proteomic platforms are restricted to a few thousand peptides, and do not allow the same level of unbiased profiling as Ribo-seq or Polysome-seq.

(ii) Perform a similar analysis in an inbred mouse strain, since rDNA CN variation exists even within an inbred mouse strain. The advantage of this approach is likely to be relatively fewer numbers (hundreds instead of thousands) as non-rDNA genetic variation is minimized, and ease of tissue access. However, it is unknown whether the results would be relevant for humans.

(iii) If we restrict ourselves to asking whether rDNA CN influences translational output, without any phenotypic assessment, then we could perform Ribo-seq or Polysome-seq in cell lines. One good option would be the Lymphoblastoid Cell Lines (LCL) resource derived from the samples included in the 1000 Genomes project. These have been fully sequenced already. For sufficient statistical power, we would need to assay all available LCLs (~2,500), and investigate if rDNA CN associates with increased protein content.

The main problem with all the above is the very high cost (>£200,000) and time investment (several years) required. These are important experiments to perform in the future, but outside the scope of the current manuscript, in our opinion.

3. “As noted by the authors the estimated rDNA CN distributions are noticeably different across sequencing centers. This is adjusted for during association testing. However, it would be helpful to see key association results, such as the significant phenome-wide association scan results in Figure 2 or the renal function results in Figure 4, stratified by sequencing center. If each is analyzed separately do the trends remain consistent?”

Response: We have now done this for all the core large-scale discovery analyses in the manuscript, showing that the trends remain consistent throughout when separating by sequencing centre. First, we provide PHESANT results from Fig. 2A separated by sequencing centre in fig. S10. Although some associations don't reach the FDR-significance threshold due to the smaller sample sizes, all retain the same trends as in the combined analysis. Please refer to the new Supplementary Tables S3-4 for a full breakdown of the results. Similarly, all nominally-significant results in Fig. 4A retain the same trend when split by sequencing centre, as shown by new fig. S14. Glomerular Filtration Rate estimates in 4B all remain negatively associated with rDNA CN when considering the sequencing centres separately. This is now presented in fig. S15B.

4. “In addition to testing each phenotype separately as in the phenome-wide scan, a separate set of analyses is performed by jointly modeling all blood count traits in a single model. I found these analyses confusing and was unsure exactly what model is being fit. For example, are blood

counts being used in a joint linear model to predict rDNA CN? That would suggest a model where blood traits influence rDNA rather than the other way around. Also, since many of these terms are co-linear the interpretation of the resulting coefficients is challenging and it was not clear what the main takeaway from those analyses is. There are techniques specifically developed for jointly testing multiple phenotypes which may be more appropriate and account for trait-trait correlation.

Response: We have now extended the description of the multivariate regression analyses in the Methods, and referred them as such in the main text, instead of the previous “joint” nomenclature, (which we agree could be confusing, even though regression models do not assume any direction of influence). Moreover, since the results of these analyses are not essential for the conclusions of the manuscript, we have now moved the top part of figures 2B and 4B to the supplementary materials (figs. S11 and S14A, respectively). In addition, the remaining previously-reported “joint” models (in Figs. 3B, 3C, and 3D, as well as previous supplementary figure S14, now S17) have been converted to conventional separate “univariate” models, with the blood counts as response and rDNA CN as explanatory variable alongside the standard covariates. No conclusions change due to these modifications.

5. The text mentions three references (7-9) but it was unclear how the current study relates to the approaches for analysis of pleiotropic loci used in those studies. The rationale behind this part should be better explained.”

Response: Apologies for the confusion. This comparison is not essential to the manuscript and we have now removed this sentence.

6. “It would be helpful to provide code and additional description of how rDNA CN was analyzed in WGS from UKB. Presumably these were analyzed on the DNA Nexus RAP, which is the only place the WGS data can currently be accessed? But the process of doing this on the cloud resource is not described. Additionally, although they cannot be publicly posted, rDNA CN estimates could be uploaded to UK Biobank to be made available to other researchers.”

Response: The Methods section now explicitly mentions the procedure used to access and obtain read counts from the WGS data within the UKB RAP. We have also included scripts in our code repository that show the explicit instructions we used to achieve this. All 18S Ratio estimates, with and without adjustment for sequencing centre, will indeed be submitted as a Return to the UKB upon acceptance of the manuscript.

Minor comments:

1. *"The authors used "hg38 consensus assembly" (line 16, page 3). Should this say instead "hg38 reference assembly"*

Response: The reviewer is correct and it has now been fixed.

2. *"The phrases "however unlikely" are often used in the manuscript and can be removed. E.g., "We wanted to address the possibility, however unlikely..."*

Response: This phrase has been removed from the manuscript.

3. *"For Figure 3E, the authors claim, "The differences in the strength of the association amongst the ethnicities may reflect biological differences, or be technical in nature due to, for example, population stratification within each group." What about reduced power from low sample sizes or different ranges of copy numbers?"*

Response: We agree and now mention these as potential reasons.

4. *"The X-axis label of Figure 2D is confusing. Not all of them are NLR. Also, it seems to be a duplicate of supp figure 11 which has a different x-axis label. Additionally, it was not clear how Figure 2D results implies no reverse causality."*

Response: Apologies, this was a typo which has now been corrected, and previous fig. S11 removed due to its redundancy. For the interpretation of Figure 2D with regards to reverse causality, please refer to our detailed response to Point 1 raised by Reviewer 2 below.

5. *Some figures show confidence intervals around the regression coefficients + p-value together, while others don't (E.g. Figure 4). It would be good to keep these consistent and show both wherever association testing results are shown.*

Response: We agree and all main figures only presenting effect sizes now include significance levels (3E, 4A, 4B, and 4D), aside from those that already included both. Similarly, fig. S13, S14, and S15 also include both.

6. *“It will be helpful to show the sample number in each bin for Fig. 2f so we know if the bins with large deviations from the others are driven by low sample size.”*

Response: We now do this. Indeed, the bins with the large deviations have sample numbers approximately 1-2 orders of magnitude smaller than the rest.

7. *“The authors used “strikingly” several times to describe the 23 blood associations identified. This may not be surprising, as those are likely the most powered traits in UKB.”*

Response: The reviewer is correct and we have toned down the text.

8. *“Previous studies suggest that individuals of African ancestry can have a higher eGFR level and requires a separate formula. Is this taken into account?”*

Response: We apologise for not clearly stating that only individuals of White British ancestry were used for this analysis. This has now been fixed in the relevant figure caption.

9. *“For Figure 3E, the Chinese population results are highly distinct from the other groups and appear to be more significant. This is surprising as the Chinese sample size is likely smaller than that for other groups. Adding sample size and significance values to 3E would be helpful.”*

Response: Figure 3E has now been amended.

10. *“In the methods, “Entries prior to 1904 or posterior to 2036 were removed”. Are these years? In which case entries later than 2023 should also be problematic?”*

Response: These dates were used as thresholds according to the description of data encoding 819 in the UKB (<https://biobank.ndph.ox.ac.uk/showcase/coding.cgi?id=819>). As no entry appears with a 1909 date, the 1904 threshold was sufficient. In reality, since only one of the entries has any date with any of those values ('1903-03-03'), we have now rewritten the Methods section to clarify accordingly. No upper bound was actually necessary (even if our code explicitly required so), thus we have avoided mentioning it in the new version to avoid confusion.

11. *“Many plots show rDNA CN on the y-axis and trait values on the y-axis. However, it is more typical to have the predictor (here, rDNA CN) on the x-axis instead. (e.g. Figure 4c and similar plots).”*

Response: We have now flipped the axes in Figs. 2D and 4C.

Reviewer #2:

1. “My primary reservation comes over the conclusion of the lack of reverse causation from inflammation. I am not convinced by this, which may be my ignorance as it is poorly described. Specifically, in the text this is only covered in half a sentence, and I don't understand the description of the relevant figure (2C) from the legend. Thus, it is not clear to me that this does demonstrate lack of reverse causality. I want to be convinced of this before I am prepared to accept the main conclusions of the work”.

More generally, the authors have used various means to try and rule out causal relationships other than rDNA CN => phenotype. While I think their conclusions are likely correct, I don't think they are able to conclude this as cleanly as they have. For example, they use lack of GWAS hits on CN to conclude there are not genetic factors that drive both CN and blood composition. However, it remains possible that genetic factors not included in the GWAS, such as structural variation, could be a causal driver of CN and composition. Similarly, there may be other causal factors, such as developmental processes, that the authors haven't checked here. I emphasize that I am not suggesting these alternative causal explanations are necessarily likely, but I do not believe the authors can formally exclude them. Therefore, this needs to be acknowledged in appropriate places in the text, particularly the abstract and discussion.

In the abstract, the authors go straight from association to causation without any description of the careful work they have done to winnow in on causation. Therefore, as currently written it sounds unscientific and is not representative of the work performed. The abstract should include description of the work performed to indicate that a causal relationship is the most likely explanation of the data (and that it is not the only explanation).”

Response: The direction of causality is indeed a critical issue that we address in our manuscript. It isn't one, but rather multiple complementary analyses that lead us to conclude it is very unlikely that the rDNA CN v phenotype associations we report are a result of environmental or genetic factors outside the rDNA loci (indeed the reviewer also considers other explanations unlikely). Nevertheless, our description in the previous version of the manuscript was certainly inadequate. We have rewritten key sections in the Results and also explain our reasoning below.

First, we find that rDNA CN is strongly associated with established blood cell-based markers of inflammation – NLR, PLR, SII (and to a lesser extent LMR). So, does inflammation lead to changes in rDNA CN, i.e., reverse causality? Typically, one would use Mendelian Randomization

in such a scenario but, given the lack of association between rDNA CN and SNVs in the rest of the genome, this isn't possible. However, all of these markers have been extensively studied in many different contexts. We therefore asked the question: if we look at a biological context where we know that NLR changes (using just one blood cell-based marker as an example) then, if reverse causality is at play, we should see a concerted change in rDNA CN (these are the analyses described in Figure 2). The association between age and NLR is well established and indeed the P value for this association in the UKB White British individuals is $P < 10^{-200}$. However, there is no association between age and rDNA CN. We then look at a range of other contexts where it has been previously conclusively established that NLR changes. This includes cancer, circadian fluctuations, and seasonal fluctuations. In no case does rDNA CN show concomitant changes.

Other analyses we present argue further against reverse causality. First, is the observation relating to C reactive protein (CRP). CRP is a very well-established marker of inflammation. Indeed, when we run a PHESANT analysis on NLR itself, the association with CRP is so strong that the P value returned by PHESANT is '0' (Table S6). However, CRP does not even cross the FDR corrected threshold in the PHESANT analysis for rDNA CN. In fact, if we account for blood composition in the rDNA CN vs CRP association, the effect size is virtually non-existent, as it is later shown in Fig. 4A. We now reference that figure in the re-written paragraph. Furthermore, when we analyse individuals without any recorded disease, and hence one would expect these individuals to not display disease-associated elevated NLR, the effect size of the rDNA CN and NLR association actually gets stronger (Fig 3C). In other words, although CRP is a very strong marker of inflammation, there is no robust association with rDNA CN, further arguing against rDNA CN variation being a result of inflammation.

Taken together, the multiple analyses presented in Figures 2 and 3 argue against reverse causality. Rather, we think rDNA CN may be subtly influencing the balance between the myeloid and lymphoid differentiation pathways in hematopoietic stem cells. This is of course just speculation at the moment, and we also agree with the reviewer that we need to be more conservative in our conclusions and consider alternative explanations. We have now toned down the claims throughout the manuscript and noted the limitations more explicitly. First, we have changed the title to reflect the fact that we have only looked in the UKB. Second, we have reworked the abstract (within the 150-word limit). Third, in the first paragraph of the discussion, we note alternative explanations and additional limitations. We hope that the revised manuscript provides a clearer, more detailed and balanced interpretation of our results.

2. *“Finally, I like controls. It would be nice to have a control indicating that significance scores from the phenome screens actually show something biological. For example, the authors could use as a screen input something like the sum of the participant ID number digits (if such a thing exists), or something else totally random. Nothing significant should show up, obviously - if it does, it would indicate some problem.”*

Response: This is a good suggestion and we now provide the results of a full PHESANT analysis done on the same samples, but we randomly permuted the sample IDs, whilst keeping all the phenotype categories the same so as to preserve the data structure. There are no hits at an FDRcorrected level. The results of this analysis are presented in the new Table S2 and detailed in the Methods.

Other comments

3. *“The Introduction is extremely short. Is this normal for Cell Genomics? If so, I guess that is fine, but otherwise it looks odd. Also, while the first sentence is not incorrect, it implies an incorrect view (that these features are specific to mammals) and so should be reworded to avoid this implication. Finally, their claim of genome studies ignoring this is not correct - there are several who have done so, some of whom have been cited by the authors here.”*

Response: We now provide a longer introduction to better contextualise our study.

4. *“On line 32 what does “adjusted estimates” mean, and similar for the S2 legend?”*

Response: When this is introduced in the main text, we have reworded it to read “rDNA CN estimates adjusted for sequencing centre” instead, which we think makes its meaning clearer. In addition, both in the text and figure captions, we now point at the Methods section, where we also make its explanation more explicit.

5. *“There is some association between time and CN in Fig2F - can the authors rule this out as a causal effect? It would be helpful in this respect to know what the N is for each of these time bands in the graph (and for 2E).”*

Response: We now include sample sizes for each bin in both figures, which shows that apparent differences only arise at times of the day where orders of magnitude fewer individuals were sampled.

6. *"I couldn't find supplementary table 3 - does this have the 15 categories talked about on line 8 of page 6? If not, these should be included somewhere."*

Response: Apologies, as this may have been a problem during manuscript submission. We have now fixed and can confirm Table S3 is present, currently as Table S6.

7. *"A paragraph shouldn't start with "To note..". I don't understand what this paragraph (on page 6) is about - where does it "dropped to virtually zero"?"*

Response: Please refer to the third paragraph of the Response to Point '1' above.

8. *"In the discussion on page 8 around line 10 on possible mechanisms, this is far from the only option and seems an unnecessarily restricted choice. I think a modification of this to either expand the possibilities or clearly indicate it is just an illustrative example is merited. Similarly, on line 15 what evidence is there that epigenetic state impacts this? I think this is strictly speculation."*

Response: We have re-worked the discussion to consider other possibilities.

9. *"In the Figure 2 legend it would be nice to write somewhere what NLR, PLR and SII mean so the reader doesn't need to go back to the text. Also, for this figure plus 3D and 4C there is no description of what the colours mean."*

Response: Explicit descriptions of the acronyms and the colour shades have now been included in the figure captions.

10. *"Why was FigS5 not done for each sequencing centre separately?"*

Response: We have now done this.

11. *"The materials and methods don't have page or line numbers, but I think it is on the bottom of page 4 - I don't understand the description of the rDNA reference here. Does this mean that actually more than 1 entire unit was used as the reference? A better description of the reference is warranted and the sequence used should be provided (e.g. as a fasta file as part of the supplementary). Underneath this, what are "singletons"?"*

Response: We now provide a more detailed description of the reference rDNA unit sequence in the Methods. Since this reference was already used and described in full detail in a previous publication (Rodriguez-Algarra, Seaborne et al, *Genome Biology*, 2022), we have summarised

the description to avoid redundancy. In any case, we now include the corresponding fasta file in the code repository. We now also include a description of what “singleton” reads mean in the Methods.

Referees' report, second round of review

Reviewer #1:

The authors have included additional analysis and revised the text to enhance the manuscript. The new evidence added gives confidence that the association signals themselves are robust and reproducible across centers and allows focusing more on the interpretation of these signals.

However, I still have several concerns regarding the interpretation. The key findings of associations of rDNA CN with blood cell counts and renal function are plausible. However, the logic the authors use to prove this is not due to either (1) reverse causality or (2) confounding are difficult to follow and I am not convinced from the paper and the response to reviewer 2 that they have proven either of those. It is true that some of the ways these associations could be proven are currently prohibitively expensive, as the reviewers describe in their response. Still, the manuscript concludes without providing concrete answers.

Major points:

1. Logic to rule out reverse (1) causality and (2) confounding:

For (1) the authors investigate whether markers of inflammation such as NLR (which are based on blood cell counts), could themselves be *causing* rDNA CN, which could lead to the observed associations. These markers and rDNA CN are indeed highly correlated, across the range of inflammation marker values. But, there can be elevated NLR (e.g. associated with age, circadian rhythm, or CRP) without a corresponding change in rDNA CN. The authors use this point to argue against reverse causality.

But this does not have to be the case. It's possible e.g. that blood cell types in the past or during development lead to changes in rDNA CN and also influence present day blood composition. Therefore these short term changes (e.g. circadian rhythm) would still be associated with NLR but not rDNA CN. It is also not clear why there is a jump directly to NLR and other inflammation markers in the discussion of reverse causality. Why not just analyze blood cell counts or the PCs derived from those directly, which is where the original signal was observed?

That being said, I actually think reverse causality is unlikely here. It's just that the arguments provided based on NLR and CRP are not convincing of this point. (the fact that rDNA CN tracks as expected across relatives, and is mostly identical in MZ twins, suggests it is pretty stable and does not fluctuate with things like short term inflammation and to me could be better argument against reverse causality).

For (2), the authors look for potential confounding factors by overlapping phewas results for rDNA CN and inflammation markers. They then used those shared phewas hits (e.g. urate,

cystatin c, others) as covariates and still find significant hits with blood biochemistry. The logic described on page 7 was a bit hard to follow and it only became clear after reexamining figure 3 what was being done. (They also do a series of increasingly stringent filters, which are good but I don't think relate to the confounding argument, which is how they are framed.)

A challenge with this section is there may be confounding factors that were not actually measured in the phewas or that the study is underpowered to detect. Similarly to (1), I don't necessarily expect that to be the case. However the current analysis still does not prove this point.

2. The section on kidney function is still very sparse

All there is really to go on in the results for the kidney function section is the reported association statistics. The authors have added proteomics analysis, which is great, potentially very interesting, and likely involved a large amount of work to get processed. However that analysis is buried in the discussion whereas it should really be in the results. It is interesting that there is an excess of positive betas. It would have been helpful to see if these trends (correlation of effect sizes, excess of positives betas) are specific to kidney function or if those trends are found for other traits analyzed associated with rDNA CN (and, as a control, *not* found in traits not associated with rDNA CN).

Minor points:

1. In the introduction, the authors mention, 'a key limitation of previous genetic association studies has been the focus on the single-copy portion of the genome, and repetitive genomic features have been largely ignored.' Tremendous effort has recently been invested in studying the association between repetitive DNA sequences and phenotypes, including VNTRs (e.g. PMIDs: 34554798, 37527660), STRs (e.g. <https://doi.org/10.1101/2022.08.01.502370>), and telomere length (e.g. PMID: 34611362). It would be worth mentioning that with the new availability of large WGS cohorts this is now changing.

2. Page 5 line 30: "It should be noted that this combination of positive and negative associations with various blood subtypes is a feature of rDNA CN, as all pairwise correlations between blood cell subtype counts in the UKB are in fact positive": can't some of this be explained by putting multiple co-linear variables into a multivariate regression?

3. Page 6 line 35: "the in the" -> "in the"

4. Figure 2E: There does actually seem to be significant month to month variation in rDNA CN?

5. Note: Figure S9: rs62153030 actually does look like there is a signal in both cohorts. It is weak individually but meets GW-sig when combined. This is exactly the type of thing that happens often in GWAS meta-analysis. Agree it doesn't look very convincing as is, but might tone down language about no genetic influences.

Reviewer #2:

The manuscript is improved but I was a bit disappointed the authors didn't do more to clarify the results in Fig2, which was my main comment. There are also a few other small outstanding issues.

I appreciate the greater clarity around the reverse causality argument and the caveats that have been added to the manuscript. However, a key point of my previous review was that I didn't understand Fig2C, and the description of this is essentially unchanged in the revised version. To be more specific, 2B shows a significant association between inflammation markers and CN, while 2C, which is said to involve "analysis of a range of different contexts", does not. I don't understand what the "range of different contexts" is - the figure shows age and cancer - is that what is meant? Two contexts hardly seems like a range. And why is there an association in 2B but not 2C - how has considering the "range" dissipated this association? I'm quite prepared to believe this is obvious to someone expert in GWAS-type analyses, but I think the target audience of both the paper and the journal is broader than just GWAS aficionados, making some better description of what was done, the logic, and some interpretation justified.

Other comments:

Page 3:26 - "unfortunately" seems entirely inappropriate here, and points to bias. If it happens that there is, in reality, no effect, this would not be remotely unfortunate.

Page 3:29 - I had trouble understanding what "methods that are scalable so as to measure rDNA genetic variation" means exactly - perhaps this could be reworded for clarity.

Introduction - this is improved, but typically the last part of this or the start of the results would indicate the aim and scope of the work done in the paper. I think it would help readability/understandability to include a couple of sentences stating these.

The addition of the control is nice, but there are different accounts of what was actually done. The response to reviewers says ID number, while the text says CN permutation. I guess either would be fine, but it would be good to be certain which was done.

Page 7:11 - I had trouble understanding "either "Blood biochemistry", diseases or medications in the UKB". Perhaps it's just missing a comma after "diseases"?

Page 8:3 - the following text seems contradictory: "on white British individuals not recorded as taking statins - as they are known to affect several biomarker measurements (25). Retaining all individuals yielded", making it unclear what was actually done. Plus, this part doesn't clearly point to the actual results. This should be reworded for clarity.

Page 9:26 - in what way does: "we were limited to measuring bulk rDNA CN as opposed to individual rDNA unit-level analysis" suggest that CN impact is underestimated? The statement is true, but it seems irrelevant for determining the impact of CN.

Authors' response to the second round of review

Major Updates**1. Reverse causality**

The lack of association between rDNA CN and genetic variation in the rest of the genome means that we have had to perform a range of analyses not typically required for genetic association studies. We therefore need to make every effort to convince readers and are grateful to the reviewers for their suggestions for improvement, and hope we provide greater clarity below. The key changes we have made are:

(i) Updated MZ twin analysis. As Reviewer 1 points out, the MZ twins are integral to the argument against reverse causality/common confounders. The MZ plot in the previous Fig. 1E didn't accurately convey just how similar the twins actually are in terms of rDNA CN. In that figure, we had used rDNA CN values that are adjusted for sequencing centre. However, such computational adjustments are never perfect. In the previous **Figure S7B (Plot A, right)** we showed a plot that considers only those MZ twin pairs for which both co-twins were sequenced at the same centre. This yielded an intra-MZ pair correlation for rDNA CN of 0.95. This is much stronger than the correlation we reported in Fig. 1E (i.e. $R = 0.84$ when we use rDNA CN values that have been computationally adjusted for sequencing centre).

We were previously hesitant to place too much emphasis on the above plots since in the original 200k WGS UKB dataset we analysed, both co-twins were sequenced at the same centre for only a small subset of the twin pairs (**Plot A** above has only 12 MZ pairs). In December 2023, the UKB released WGS data for the remaining 300k samples. We processed the new MZ twin data and confirm a very high intra-pair rDNA CN correlation ($R = 0.95$, **Plot B**, $n = 41$ different MZ twin pairs from the new WGS data. Note – there is no 'Sanger Vanguard' in the new 300k data as that was just the pilot phase of the UKB WGS project).

The fact that rDNA CN can potentially take any value from ~200 to 400 copies, and that these measurements almost certainly still include some technical noise, an R of 0.95 underlines just how similar MZ twins are in terms of rDNA CN. The new WGS data now gives us confidence to say that the MZ twin analysis does indeed provide strong evidence for the somatic stability of rDNA CN. In the new version of the main text, we include the relative comparisons within sequencing centre batches for both first and second WGS release.

(ii) Figure 2 and associated text. We have completely rewritten this section (starts line 12, pg 6). First, we use the example of mitochondrial copy number variation (mtDNA CN) to explain what we mean by reverse causality (**please refer to the new Fig. 2C in the manuscript**). It has been shown for multi-copy elements such as mitochondria and telomeres that copy number can vary by blood subset type. So, apparent inter-individual variation in mtDNA CN, for example, in

whole blood could be due to different levels of neutrophils amongst the individuals. Given that the UKB WGS data is derived from whole blood DNA, we had to consider a similar case of reverse causality might exist for rDNA CN, i.e. some blood cell subtypes harbour different numbers of rDNA copies compared to other blood subtypes. (Note – the below arguments hold even if multiple blood cell subtypes types harbour more copies of rDNA relative to other blood cell types)

It well established that neutrophil counts and NLR increase in many different physiological contexts. Based on published literature, the previous Figure 2C has been expanded to include other common contexts where baseline levels of NLR are significantly higher (new **Figure 2D and its extended version in fig. S13**, reproduced below). Some of the previous confusion regarding this figure may have arisen from the possibility that the yaxis could be misinterpreted as only representing associations with NLR. This is not the case, as the y-axis is simply p-value of any of the associations we are comparing. The white/grey bars show the association between physiological contexts (e.g., age) with NLR, and the black bars shows the association between the same physiological context e.g., age and rDNA CN. We are not directly associating rDNA CN with NLR in the above figure, as that was already established in Fig. 2B.

We will use age as an example. Age is very strongly associated with NLR in the UKB [when considering all individuals in UKB (white bar), just White British (WB, light grey), or just those WB individuals for which we also have WGS data (dark grey)]. Now, theoretically, if rDNA CN

varies amongst blood subtypes, e.g., higher in neutrophils compared to other blood subsets, and the rDNA CN value we measure in whole blood is simply a consequence of NLR, i.e., reverse causality, then age *should* show a significant association with rDNA CN. The black bars represent the results of the association analysis for rDNA CN with sex, age, cancer, HIV status, or Type 2 Diabetes, and there is no statistically significant association in any of the cases. Two other contexts where known fluctuations of NLR has no bearing on measured rDNA CN are shown in Figs 2E (circadian) and F (season) in the manuscript. Therefore, the association of rDNA with NLR is not influenced by known drivers of NLR that we considered. It is also worth comparing previously published MZ twin or ageing correlations for mtDNA CN and telomere length with those for rDNA CN.

(A) Xing et al., *J Natl Cancer Inst* (2008). Not based on UK Biobank data. (B) Bischoff et al., *Twin Res. and Human Genet.* (2005). Not based on UK Biobank data. (C) MZ twin rDNA CN data from the second WGS release. (D) Gupta et al., *Nature* (2023). Based on UK Biobank data. (E) Vaiserman and Krasnienkov, *Front. in Genet.* (2021). Not based on UK Biobank data. (F) Our data. rDNA CN shows no correlation with age (40 – 70 years) (Supp. Figure 4).

To summarise the above arguments in general terms, we observe a correlation between A and B, but initially don't know the directionality of this correlation.

We then look in cases where we know B changes significantly. However, in none of these cases do we see A changing. Therefore, B cannot influence A.

We hope that the explanation above clarifies why blood composition has no bearing on measured rDNA CN, and reverse causality is not involved in the association between rDNA CN and NLR.

2. Common confounder. We have also rewritten and expanded this section (starts line 23, pg 7). We highlight how MZ twins are also very useful for arguing against confounding, and why early life factors are unlikely. Nevertheless, just to be on the safe side, we still note caveats in the Results and Discussion

3. Proteomics

"The authors have added proteomics analysis, which is great, potentially very interesting, and likely involved a large amount of work to get processed. However that analysis is buried in the discussion whereas it should really be in the results. It is interesting that there is an excess of positive betas. It would have been helpful to see if these trends (correlation of effect sizes, excess of positives betas) are specific to kidney function or if those trends are found for other traits

*analyzed associated with rDNA CN (and, as a control, *not* found in traits not associated with rDNA CN)."*

We have now substantially expanded the proteomics analysis which appears in the main Results. Please refer to the new subsection "Analysis of rDNA CN-associated serum proteomic profiles" on Page 9.

We performed all the analyses requested by the reviewer and provide a detailed explanation of what was done. This includes the rationale behind the analysis, which phenotypes we focus on, how we chose an appropriate control, the results of the comparisons of proteomics profiles between the phenotypes of interest and rDNA-associated proteomic profiles, and the analysis of betas. We also present relevant caveats.

4. Analysis of the new ~300k UKB WGS data

We now present an analysis of the new ~300k WGS dataset released by the UKB in December, 2023 (starts line 3, pg 11). We have taken the conservative approach and focussed on the "Blood Count" and "Blood Biochemistry" traits that were called as significant at an FDR of < 0.01 in the first release dataset, and form the core of the manuscript. We find that these core trait associations are also significant in the second release data. The association with kidney disease was also recapitulated.

Importantly, by analysing the new ~300k data separately, we also found that a GWAS of rDNA CN in the new data did not replicate the single variant (rs62153030) that was associated with rDNA CN at a whole-genome significance level ($p = 3.4 \times 10^{-8}$) in the first release dataset. This further emphasises the false positive nature of this sole variant, confirming that rDNA CN is not influenced by genetic variation elsewhere in the genome.

We also hope the reviewers see that it isn't realistic for us to start from scratch again, perform a brand-new discovery analysis using all the 500k data, and then rewrite the entire manuscript at this advanced stage of the review process.

- (i) This would take an inordinate amount of time and funds, and the manuscript would have to go through further multiple rounds of review and revisions.
- (ii) Any new associations will have to be analysed and presented in detail, which would involve us having to first identify, and then have extensive discussions with domain specific experts for each trait.
- (iii) The increase in statistical power would not actually be that significant, given that we would be approximately doubling the dataset, not, for example, increasing it by an order of magnitude.

The next stage for future analyses should be to perform a studies that leverages not just the entire UKB data, but also other WGS datasets, especially of other ancestries, to build on the findings reported here. We will of course deposit rDNA CN values for all ~500k samples with the UKB for others to follow up.

Minor/Other Points

Reviewer 1

1. In the introduction, the authors mention, 'a key limitation of previous genetic association studies has been the focus on the single-copy portion of the genome, and repetitive genomic features have been largely ignored.' Tremendous effort has recently been invested in studying the association between repetitive DNA sequences and phenotypes, including VNTRs (e.g. PMIDs: 34554798, 37527660), STRs (e.g. <https://doi.org/10.1101/2022.08.01.502370>), and telomere length (e.g. PMID: 34611362). It would be worth mentioning that with the new availability of large WGS cohorts this is now changing.

> This has now been done. Please refer to pg 1, lines 9-11.

2. Page 5 line 30: "It should be noted that this combination of positive and negative associations with various blood subtypes is a feature of rDNA CN, as all pairwise correlations between blood cell subtype counts in the UKB are in fact positive": can't some of this be explained by putting multiple co-linear variables into a multivariate regression?

> Apologies, but we didn't understand this query. To clarify, we included this statement to emphasise that the association of rDNA CN with the blood subsets doesn't just reflect the overall background relationships amongst the different blood subsets. For example, in the overall UKB dataset, neutrophil counts are positively associated with lymphocyte counts. However, rDNA CN associates positively with neutrophils and negatively with lymphocytes, even when the corresponding models are fit separately, such as in the PHESANT results in Figure 2A. This is what led us to focus on the neutrophil-to-lymphocyte (NLR) and other ratios.

3. Page 6 line 35: "the in the" -> "in the"

> This has been fixed.

4. Figure 2E: There does actually seem to be significant month to month variation in rDNA CN?

> The apparent monthly fluctuations in rDNA CN estimates are likely random noise (ANOVA $p = 0.3$).

5. Note: Figure S9: rs62153030 actually does look like there is a signal in both cohorts. It is weak individually but meets GW-sig when combined. This is exactly the type of thing that happens often in GWAS meta-analysis. Agree it doesn't look very convincing as is, but might tone down language about no genetic influences.

> A GWAS of rDNA CN in the new ~300k WGS dataset did not replicate this SNV, further arguing that this sole association was a false positive.

Reviewer 2

Page 3:26 - "unfortunately" seems entirely inappropriate here, and points to bias. If it happens that there is, in reality, no effect, this would not be remotely unfortunate.

> This has been removed.

Page 3:29 - I had trouble understanding what "methods that are scalable so as to measure rDNA genetic variation" means exactly - perhaps this could be reworded for clarity.

> This has been reworded (pg 3, lines 31-33).

Introduction - this is improved, but typically the last part of this or the start of the results would indicate the aim and scope of the work done in the paper. I think it would help readability/understandability to include a couple of sentences stating these.

> We agree and have added a sentence at the end of the Introduction.

The addition of the control is nice, but there are different accounts of what was actually done. The response to reviewers says ID number, while the text says CN permutation. I guess either would be fine, but it would be good to be certain which was done.

> Apologies for the confusion. They intend to convey the same procedure. The permutation altered the rDNA CN assigned to each sample ID, therefore disturbing which phenotype values are associated with which rDNA CN estimates. We have reworded this to make it clearer (Pg 25, line 5).

Page 7:11 - I had trouble understanding "either "Blood biochemistry", diseases or medications in the UKB". Perhaps it's just missing a comma after "diseases"?

> Apologies, the comma was indeed missing and has now been added.

Page 8:3 - the following text seems contradictory: "on white British individuals not recorded as taking statins - as they are known to affect several biomarker measurements (25). Retaining all individuals yielded", making it unclear what was actually done. Plus, this part doesn't clearly point to the actual results. This should be reworded for clarity.

> We agree and have reworded this (Pg 8, lines 31-35).

Page 9:26 - in what way does: "we were limited to measuring bulk rDNA CN as opposed to individual rDNA unit-level analysis" suggest that CN impact is underestimated? The statement is true, but it seems irrelevant for determining the impact of CN.

> This statement has been removed.

Referees' report, third round of review

Reviewer #1:

The authors have significantly improved the manuscript and my remaining concerns have been addressed.

The new data on MZ twins provides further evidence of somatic stability of rDNA CN. The rationale behind the reverse causality analyses is also now much more clear.

I certainly appreciate how hard it is to analyze the 500K UKB data! I agree with the authors that is not necessary for this study.

One more note: although the individual-level data cannot be shared publicly, it should be possible to share the rDNA calls with UKB to distribute to approved users so they could be incorporated into future studies by other groups.

Reviewer #2:

I think the manuscript is now almost ready for publication. There are three minor wording changes I suggest to the authors to make.

1. In the last sentence of the summary it says "a novel source of genetic variation". This is not true - rDNA CN variation is a well-known source of variation. Possibly the point is semantic - the meaning of "genetic variation" - but I would think the authors want to suggest here instead that rDNA CN variation is a source of blood phenotype variation, and I invite them to do so
2. In the 1st paragraph of the introduction, I suggest that rewording to "of most previous genetic association studies is that repetitive/multi-copy features have been ignored." would be a better way of phrasing.
3. In the last sentence of page 12 starting "Finally", this is not an underestimate reason. I think the sentence should either be deleted or rephrased to represent what its true meaning is

Otherwise, I have just one issue with the manuscript, which is the paragraph on page 8 attempting to rule out common confounders. I don't buy this argument. Imagine that CN is inherited and is causal for blood phenotype. MZ twins are expected to have the same CN (inheritance) and same blood phenotypes (causation). Let us instead imagine that CN and blood phenotype can both be independently influenced by a non-genetic confounder. A MZ twin pair is expected then to have the same CN and the same blood phenotype (due to confounder). In other words, we cannot distinguish between the two. The authors' arguments rest on differences, but the point is MZ twins will usually be the same.

I don't know why the authors are resistant to the possibility of a confounder - it doesn't weaken the manuscript, and not being able to rule out a confounder is not the same as saying the effect is due to a confounder. Indeed, I would agree with the authors that most likely it is a causal relationship. Leaving the possibility of a common confounder would require changing Figure 5F, but the authors could even consider removing this panel - it is pretty obvious from a text description alone.

Authors' response to the third round of review**Reviewer #1:**

The authors have significantly improved the manuscript and my remaining concerns have been addressed.

The new data on MZ twins provides further evidence of somatic stability of rDNA CN. The rationale behind the reverse causality analyses is also now much more clear.

I certainly appreciate how hard it is to analyze the 500K UKB data! I agree with the authors that is not necessary for this study.

One more note: although the individual-level data cannot be shared publicly, it should be possible to share the rDNA calls with UKB to distribute to approved users so they could be incorporated into future studies by other groups.

> We thank the reviewer for the positive feedback. We will indeed deposit all rDNA CN calls with UK Biobank under project ID 83271.

Reviewer #2:

"I think the manuscript is now almost ready for publication. There are three minor wording changes I suggest to the authors to make."

> We are very pleased that the reviewer thinks the manuscript is almost ready for publication.

"1. In the last sentence of the summary it says "a novel source of genetic variation". This is not true - rDNA CN variation is a well-known source of variation. Possibly the point is semantic - the meaning of "genetic variation" - but I would think the authors want to suggest here instead that rDNA CN variation is a source of blood phenotype variation, and I invite them to do so."

> We have toned down the last sentence and changed it to: *"Our work demonstrates that rDNA CN is a genetic influence on trait variance in humans."*

"2. In the 1st paragraph of the introduction, I suggest that rewording to "of most previous genetic association studies is that repetitive/multi-copy features have been ignored." would be a better way of phrasing."

> This has now been amended as suggested (line 7, pg 3).

"3. In the last sentence of page 12 starting "Finally", this is not an underestimate reason. I think the sentence should either be deleted or rephrased to represent what its true meaning is."

> This sentence has now been re-written as follows: *"Finally, discovery analyses were conducted solely on the first WGS release, limiting the statistical power to identify rDNA-associated traits."*

"Otherwise, I have just one issue with the manuscript, which is the paragraph on page 8 attempting to rule out common confounders. I don't buy this argument. Imagine that CN is inherited and is causal for blood phenotype. MZ twins are expected to have the same CN (inheritance) and same blood phenotypes (causation). Let us instead imagine that CN and blood phenotype can both be independently influenced by a non-genetic confounder. A MZ twin pair is expected then to have the same CN and the same blood phenotype (due to confounder). In other words, we cannot distinguish between the two. The authors' arguments rest on differences, but the point is MZ twins will usually be the same."

I don't know why the authors are resistant to the possibility of a confounder - it doesn't weaken the manuscript, and not being able to rule out a confounder is not the same as saying the effect is due to a confounder. Indeed, I would agree with the authors that most likely it is a causal relationship. Leaving the possibility of a common confounder would require changing Figure 5F, but the authors could even consider removing this panel - it is pretty obvious from a text description alone."

> Apologies if there has been any misunderstanding as we don't disagree with reviewer at all. Firstly, the scenario the reviewer describes above is essentially the same as one of the two possibilities we presented in our previous submission (line 14, pg 8). Specifically, we said the following: *"(ii) after the twinning event, and the rDNA CN simultaneously changes genetically in the two different foetuses. Whatever mechanism is involved in this case, it would have to be extremely finely-tuned to genetically change the rDNA CN in the same direction to an almost identical extent in most, if not all, cells in both individuals. In both cases, this factor would also need to influence adult phenotypes separately"*. We agree with the reviewer that noting the possibility of a confounder, even if we and the reviewer think it is unlikely, doesn't weaken the

manuscript. Indeed, we previously included a caveat in the second last paragraph of the 'Discussion', "*There are two key limitations to our work: (i) although we find no evidence for reverse causality or common confounders influencing our results, there could still be common factors influencing both rDNA CN and phenotype but are not detectable in the UKB data; and (ii).....*".

Nevertheless, we have toned down the text further. The last sentence of the 'Results' sub-section titled "The association between rDNA CN and blood cell composition is not due to reverse causality or confounder effects" previously said:

"We consider both these alternative scenarios to be highly unlikely. Therefore, collectively, we find no evidence for a confounder, or reverse causality, involved in the association between rDNA CN and blood cell subset counts."

The above has been changed to:

"We consider both these alternative scenarios to be unlikely and, collectively, our data do not support the existence of a confounder in the association between rDNA CN and blood cell subset counts. Nevertheless, detailed mechanistic studies could be carried out in the future to further probe the issue of potential confounding, if it exists."

With regards to **Figure 5F**, we think this schematic serves a useful purpose in presenting a model based on our current thinking about how rDNA CN might influence phenotypic outcomes. For this reason, we would prefer to leave it in, but do think that the associated caption can be toned down. The previous caption read:

"(F) Schematic representation of the potential (i) direct or (ii) indirect causal influences of rDNA CN on human phenotypes, such as NLR, after reverse causality and confounding have been ruled out." The new caption reads:

"(F) Schematic representation of the potential (i) direct or (ii) indirect causal influences of rDNA CN on human phenotypes, such as NLR. The red crosses indicate that our data do not support the involvement of either reverse causality or confounding effects."

Referees' report, fourth round of review

Reviewer #2:

I think the manuscript is now ready for publication.

Authors' response to the fourth round of review

NA